# Maximizing Global Model Appeal in Federated Learning

**Yae Jee Cho**                                                    *yaejeec@andrew.cmu.edu*
*Carnegie Mellon University*

**Divyansh Jhunjhunwala**                                          *djhunjhu@andrew.cmu.edu*
*Carnegie Mellon University*

**Tian Li**                                                        *litian@andrew.cmu.edu*
*Carnegie Mellon University*
*University of Chicago*

**Virginia Smith**                                                 *smithv@cmu.edu*
*Carnegie Mellon University*

**Gauri Joshi**                                                    *gaurij@andrew.cmu.edu*
*Carnegie Mellon University*

**Reviewed on OpenReview:** *https://openreview.net/forum?id=8GI1SXqJBk*

## Abstract

Federated learning (FL) aims to collaboratively train a global model using local data from a network of clients. To warrant collaborative training, each federated client may expect the resulting global model to satisfy some individual requirement, such as achieving a certain loss threshold on their local data. However, in real FL scenarios, the global model may not satisfy the requirements of all clients in the network due to the data heterogeneity across clients. In this work, we explore the problem of *global model appeal* in FL, which we define as *the total number of clients* that find that the global model satisfies their individual requirements. We discover that global models trained using traditional FL approaches can result in a significant number of clients unsatisfied with the model based on their local requirements. As a consequence, we show that global model appeal can directly impact how clients participate in training and how the model performs on new clients at inference time. Our work proposes MAXFL, which maximizes the number of clients that find the global model appealing. MAXFL achieves a 22-40% and 18-50% improvement in the test accuracy of training clients and (unseen) test clients respectively, compared to a wide range of FL approaches that tackle data heterogeneity, aim to incentivize clients, and learn personalized/fair models.

## 1 Introduction

Federated learning (FL) is a distributed learning framework that considers training a machine learning model using a network of clients (e.g., mobile phones, hospitals), without directly sharing client data with a central server (McMahan et al., 2017). FL is typically performed by aggregating clients' updates over multiple communication rounds to produce a global model (Kairouz et al., 2019). In turn, each client may have its own requirement that it expects to be met by the resulting global model under different settings such as at inference or with some fine-tuning. For example, clients such as hospitals or edge devices may expect that the global model performs *at least* better than a local model trained in isolation on the client's limited local data

before contributing to FL training. Unfortunately, due to data heterogeneity across the clients, the global model may fail to meet the requirements of all clients (Yu et al., 2020).

Previously, a plethora of works in FL on techniques such as variance reduction (Karimireddy et al., 2019), personalization (Fallah et al., 2020; Dinh et al., 2020), and fairness (Li et al., 2019) have proposed to train a global model or several personalized models that can better cater to the needs of the clients or the server. However, prior work has not directly focused on *the total number of clients* that are satisfied with *the single global model* based on their individual requirements, and has not explored how this may affect the training of the global model from the server's perspective when clients have the autonomy to freely join or leave the federation. Recent closely related works focus on clients' incentives from a game-theoretic lens (Hu et al., 2023; Kang et al., 2019; Zhang et al., 2021) and establish useful insights for simple linear tasks, but it is difficult to extend these to practical non-convex machine learning problems. Other related works design strategies specifically to prevent client dropout (Wang & Xu, 2022; Gu et al., 2021), but these algorithms are stateful, i.e., they require saving the gradient information from previously participating clients for the current updates, making them impractical to implement in cross-device settings (Kairouz et al., 2019). Moreover, these works only provide convergence guarantees to the global minimum with respect to the standard FedAvg objective (McMahan et al., 2017), lacking theoretical justification that their objectives can yield solutions that guarantee more participating clients compared to other classic FL objectives, even in simplified settings; we provide such guarantees for our proposed objective for mean estimation problems, which helps to shed light on our strong empirical performance (Section 2.2).

Proposing a new and formal metric to evaluate FL systems, we define that a global model is *appealing* to a client if it satisfies the client's specified requirement, such as incurring at most some max training loss. Subsequently, we define the number of clients which find the global model appealing as *global model appeal* (GM-APPEAL; formalized in Definition 1). We show that having a high global model appeal is critical for the server to maintain a large pool of clients to select from for training, and for gathering additional willingly participating clients. This is especially true in the light of clients possibly opting out of FL due to the significant costs associated with training (e.g., computational overhead, privacy risks, logistical challenges), which is a practical concern not typically considered in prior work. With a larger pool of clients to select from, a server can not only improve privacy-utility trade-offs (McMahan et al., 2018), but can also improve test accuracy on participating clients, and produce a global model that generalizes better at inference to new unseen clients (see Fig. 1 and Table 1).

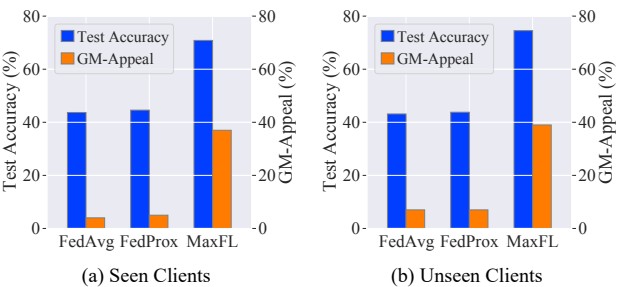

(a) Seen Clients    (b) Unseen Clients

Figure 1: Test acc. and GM-Appeal of the global model for FMNIST. A higher GM-Appeal results in a higher test accuracy for both the seen clients that have participated during training and unseen clients that have not, due to the server having a larger pool of clients to select from. MAXFL, which aims to maximize GM-Appeal, results in the highest test accuracy compared to the other baselines that do not consider GM-Appeal.

In this work, we seek to understand: (1) What benefits exist when maximizing global model appeal relative to other common federated modeling approaches, and (2) What strategies exist to maximize global model appeal in federated settings. Our key contributions are summarized as follows:

- We introduce the notion of *global model appeal* (referred to as GM-APPEAL)—the fraction of clients that have their local requirements met by the global model. We then propose MAXFL, an objective that directly maximizes global model appeal, and show that having a high global model appeal can lead to better test accuracy on training clients, as well as better generalization to unseen clients at inference time.

- We theoretically show for mean estimation that the MAXFL objective yields a solution that guarantees higher GM-APPEAL than standard FL objectives and provide convergence guarantees for our MAXFL solver which allows partial client participation, is applicable to non-convex objectives, and is stateless (does not require clients to maintain local parameters during training).

- We empirically evaluate the performance of MAXFL to thoroughly understand the benefits of maximizing GM-APPEAL with experiments where i) clients can flexibly opt-out of training, ii) there are new incoming (unseen) clients, and iii) clients perform personalization via local fine-tuning with the global model.

- We show that MAXFL significantly improves the global model appeal in practice, leading to a 22-40% and 18-50% test accuracy improvement for the seen clients and unseen clients respectively, compared to a wide range of FL methods, including those that tackle data heterogeneity, aim for variance reduction or incentivizing clients, or provide personalization or fairness.

Overall, our goal in comparing MAXFL with a variety of other FL methods that have varying goals is not necessarily to compete against these methods, but rather to understand and demonstrate the potential benefits of our proposed notion of global model appeal relative to other objectives under different scenarios, such as with flexible client participation or new incoming clients. As global model appeal has not been studied previously in FL, our work is the first to explore its possible implications, and then propose an objective to train a global model that can maximize the number of clients whose individual requirements are satisfied. We hope our proposed perspective of viewing and evaluating FL systems can inspire future works in different FL scenarios and applications where client participation is not necessarily taken for granted, and can potentially be used in conjunction with prior approaches in FL (e.g., see Section D.2). We provide a more detailed review of prior work and related areas of fairness, personalization and client incentives in Section 4.

## 2 Problem Formulation

**Setting.** We consider a setup where $M$ clients are connected to a central server to collaboratively train a global model. For each client $k \in [M]$, its true loss function is given by $f_k(\mathbf{w}) = \mathbb{E}_{\xi \sim \mathcal{D}_k}[\ell(\mathbf{w}, \xi)]$ where $\mathcal{D}_k$ is the true data distribution of client $k$, and $\ell(\mathbf{w}, \xi)$ is the composite loss function for the model $\mathbf{w} \in \mathbb{R}^d$ for data sample $\xi$. In practice, each client only has access to its local training dataset $\mathcal{B}_k$ with $|\mathcal{B}_k| = N_k$ data samples sampled from $\mathcal{D}_k$. Client $k$'s empirical loss function is $F_k(\mathbf{w}) = \frac{1}{|\mathcal{B}_k|} \sum_{\xi \in \mathcal{B}_k} \ell(\mathbf{w}, \xi)$. While some of the take-aways of our work (e.g., improving performance on unseen clients) may be more specific to cross-device applications, our general setup and method are applicable to both cross-device and cross-silo FL.

**Defining Global Model Appeal.** Each client's natural aim is to find a model that minimizes its true local loss $f_k(\mathbf{w})$. Clients can have different thresholds of how small this loss should be, and we denote such self-defined threshold for each client as $\rho_k$, $k \in [M]$. For instance, each client can perform solo training on its local dataset $\mathcal{B}_k$ to obtain an approximate local model $\widehat{\mathbf{w}}_k$ and have its threshold to be the true loss from this local model, i.e., $\rho_k = f_k(\widehat{\mathbf{w}}_k)$[1]. Based on these client requirements, we provide the formal definition of global model appeal below:

**Definition 1** (Global Model Appeal). *A global model $\mathbf{w}$ is said to be appealing to client $k \in [M]$ if $f_k(\mathbf{w}) < \rho_k$, i.e., the global model $\mathbf{w}$ yields a smaller local true loss than the self-defined threshold of the client. Accordingly, we define the fraction of clients to which the global model is appealing as global model appeal (GM-APPEAL) with $\mathbb{I}$ being the indicator function:*

$$GM\text{-}APPEAL = \frac{1}{M} \sum_{k=1}^{M} \mathbb{I}\{f_k(\mathbf{w}) < \rho_k\} \tag{1}$$

Our GM-APPEAL metric measures the exact *fraction of clients* that find the global model appealing by focusing on whether the global model satisfies the clients' requirements or not instead of looking at the gap between $f_k(\mathbf{w})$ and $\rho_k$. Another variation of (1) could be to measure the margin $\sum_k \max\{\rho_k - f_k(\mathbf{w}), 0\}$, but this does not capture the motivation behind our work which is to understand how the number of clients that find the global model appealing affects the global model performance.

**Why Explore GM-Appeal in FL?** GM-APPEAL measures how many clients have their requirements satisfied by the global model. Thus, it can gauge important characteristics of the server's global model such as i) how many clients are likely to dropout with the current global model or ii) how many new incoming

---

[1]The client can have held-out data used for calculating the true loss $f_k(\cdot)$ or use its training data as a proxy. We explain in more detail of defining $\rho_k$ in Section 3.

clients that do not have the capacity for additional training will likely be satisfied with the current global model without any additional training to the model. Ultimately, a high global model appeal can lead to a larger pool of clients for the server to select from. The standard FL objective (McMahan et al., 2017) or its popular variants (Karimireddy et al., 2019; Li et al., 2020; Fallah et al., 2020) does not consider whether the global model satisfies the clients' requirements, and implicitly assumes that the server will have a large number of clients to select from. However, this may not necessarily be true if clients are allowed to dropout when they find the global model unappealing. We show that acquiring a larger pool of clients by improving global model appeal is useful for improving the global model for both the seen clients at training as well as the unseen clients at inference (see Fig. 1 and Table 1). In fact, we find that other baselines such as those that aim to tackle data heterogeneity, improve fairness, or provide personalization have low GM-Appeal, leading to a large number of clients opting out. Due to this, the global model is trained on just a few limited data points, resulting in poor performance. Our work explores a new notion of global model appeal, showing its significance in FL.

## 2.1 Proposed MaxFL Objective

In this section, we first introduce MAXFL whose aim is to train a global model that maximizes GM-APPEAL. A naïve approach can be to directly maximize GM-APPEAL defined in (1) as follows:

$$\operatorname*{argmax}_{\mathbf{w}} \text{GM-APPEAL} = \operatorname*{argmin}_{\mathbf{w}} \sum_{k=1}^{M} \text{sign}(f_k(\mathbf{w}) - \rho_k). \tag{2}$$

where $\text{sign}(x) = 1$ if $x \geq 0$ and 0 otherwise. There are two immediate difficulties in minimizing (2). First, clients may not know their true data distribution $\mathcal{D}_k$ to compute $f_k(\mathbf{w}) - \rho_k$. Second, the sign function makes the objective nondifferentiable and limits the use of common gradient-based methods. We resolve these issues by proposing a "proxy" for (2) with the following relaxations.

**i) Replacing the Sign function with the Sigmoid function $\sigma(\cdot)$:** Replacing the non-differentiable 0-1 loss with a smooth differentiable loss is a standard tool used in optimization (Nguyen & Sanner, 2013; Masnadi-shirazi & Vasconcelos, 2008). Given the many candidates (e.g. hinge loss, ReLU, sigmoid), we find that using the sigmoid function is essential for our objective to faithfully approximate the true objective in (2). We further discuss the theoretical implications of using the sigmoid loss in Section 2.2.

**ii) Replacing $f_k(\mathbf{w})$ with $F_k(\mathbf{w})$:** As clients do not have access to their true distribution $\mathcal{D}_k$ to compute $f_k(\cdot)$ we propose to use an empirical estimate $\sigma(F_k(\mathbf{w}) - \rho_k)$. This is again similar to what is done in standard FL where we minimize $F_k(\mathbf{w})$ instead of $f_k(\mathbf{w})$ at client $k$. Note that the global model $\mathbf{w}$ is trained on the data of all clients, making it unlikely to overfit to the local data of any particular client, leading to $f_k(\mathbf{w}) \approx F_k(\mathbf{w})$, which we also show empirically in Appendix D.2.

With the two relaxations above, we present our proposed MAXFL objective:

$$\text{MAXFL Obj.:} \quad \min_{\mathbf{w}} \widetilde{F}(\mathbf{w}) = \min_{\mathbf{w}} \frac{1}{M} \sum_{i=1}^{M} \widetilde{F}_i(\mathbf{w}), \text{ where } \widetilde{F}_i(\mathbf{w}) := \sigma(F_i(\mathbf{w}) - \rho_i). \tag{3}$$

Before presenting our proposed solver for the MAXFL objective, we first present a motivating toy example with mean estimation which shows that MAXFL's objective leads to a different solution that has higher GM-APPEAL compared to the solution obtained from the classic FL objective.

## 2.2 Toy Example: Maximizing GM-Appeal in Mean Estimation

We consider a toy setup with $M = 2$ clients where the true loss function at each client is given by $f_k(w) = (w - \theta_k)^2$. In practice, clients only have $N_k$ samples drawn from the distribution given by $e_{k,j} \sim \mathcal{N}(\theta_k, \nu^2), \ \forall j \in [N_k]$. We further assume that the empirical loss function at each client is given by $F_k(w) = (w - \widehat{\theta}_k)^2 + (\widehat{\theta}_k - \theta_k)^2$ where $\widehat{\theta}_k$ is the empirical mean, $\widehat{\theta}_k = \frac{1}{|\mathcal{B}_k|} \sum_{j=1}^{N_k} e_{k,j}$. It is easy to see that the minimizer of $F_k(w)$ is the empirical mean $\widehat{\theta}_k$. Thus, we set the solo-trained model at each client as $\widehat{w}_k = \widehat{\theta}_k$ and the loss threshold requirement at a client as $\rho_k = F_k(\widehat{w}_k) = (\widehat{\theta}_k - \theta_k)^2$.

**GM-Appeal for Standard FL Model Decreases Exponentially with Heterogeneity.** For simplicity let us assume $N_1 = N_2 = N$ where $\gamma^2 = \nu^2/N$ is the variance of the local empirical means and $\gamma_G^2 = ((\theta_1 - \theta_2)/2)^2 > 0$ is the measure of heterogeneity between the true means. The standard FL objective will always set the FL model to be the average of the local empirical means (i.e. $w = (\widehat{\theta}_1 + \widehat{\theta}_2)/2$) and does not take into account the heterogeneity among the clients. As a result, the GM-Appeal of the global model decreases *exponentially* as $\gamma_G^2$ increases.

**Lemma 2.1.** *The expected GM-Appeal of the standard FL model is upper bounded by* $2\exp\left(-\gamma_G^2/(5\gamma^2)\right)$, *where the expectation is taken over the randomness in the local datasets $\mathcal{B}_1, \mathcal{B}_2$.*

**Maximizing GM-Appeal with Relaxed Objective.** We now maximize the GM-Appeal for this setting by solving a relaxed version of the objective in (2) as proposed earlier. We replace the true loss $f_k(\cdot)$ with the empirical loss $F_k(\cdot)$ and replace the 0-1 (sign) loss with a differentiable approximation $h(\cdot)$. We first show that setting $h(\cdot)$ to be a standard convex surrogate for the 0-1 loss (e.g. log loss, exponential loss, ReLU) leads to our new objective behaving the same as the standard FL objective.

**Lemma 2.2.** *Let $h$ be any function that is convex, twice differentiable, and strictly increasing in $[0, \infty)$. Then our relaxed objective is strictly convex and has a unique minimizer at $w^* = (\widehat{\theta}_1 + \widehat{\theta}_2)/2$.*

**MaxFL Objective Leads to Increased GM-Appeal.** Based on Lemma 2.2, we see that we need nonconvexity in $h(\cdot)$ for the objective to behave differently than standard FL. We set $h(x) = \sigma(x) = \exp(x)/1 + \exp(x)$, as proposed in our MaxFL objective in (3) and find that the MaxFL objective *adapts* to the empirical heterogeneity parameter $\widehat{\gamma}_G^2 = (\widehat{\theta}_2 - \widehat{\theta}_1/2)^2$. If $\widehat{\gamma}_G^2 < 1$ (small data heterogeneity), the objective encourages collaboration by setting the global model to be the average of the local models. Conversely, if $\widehat{\gamma}_G^2 > 2$ (large data heterogeneity), the objective encourages *separation* by setting the global model close to the local model of either client (see Fig. 2 below). Based on this observation, we have the following theorem.

**Theorem 2.1.** *Let $w$ be a local minima of the MaxFL objective. The expected GM-Appeal using $w$ is lower bounded by* $\exp\left(-\gamma^{-2}\right)/16$ *where the expectation is over the local datasets $\mathcal{B}_1, \mathcal{B}_2$.*

Observe that even with $\gamma_G^2 \gg 0$, MaxFL will keep satisfying the requirement of at least one client by adapting its objective accordingly. We show the behavior of MaxFL in a 3-client setup which further highlights the non-trivialness of our proposed MaxFL's formulation in Appendix A along with the simulation details for Fig. 2. Details of our proof in this section can be found in Appendix B.

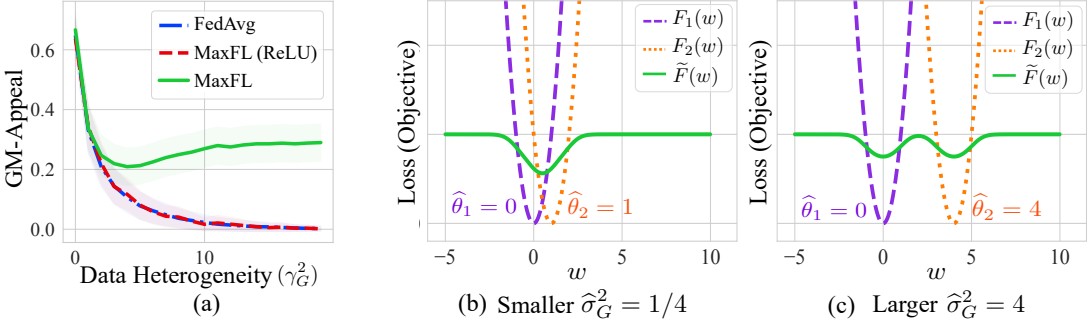

Figure 2: (a): GM-APPEAL for FedAvg decays exponentially while GM-Appeal for MaxFL is lower bounded. Replacing the sigmoid approximation with ReLU in MaxFL leads to the same solution as FedAvg. (b-c): MaxFL adapts to the heterogeneity of the problem—for small heterogeneity it encourages collaboration by having a single global minima, for large heterogeneity it encourages separation by having far away local minimas.

## 3  Proposed MaxFL Solver

In this section, we present our MaxFL objective's solver. The MaxFL algorithm enjoys the following properties: i) uses the same local SGD procedure as in standard FedAvg, ii) allows partial client participation,

and iii) is stateless. By stateless, we mean that clients do not carry varying local parameters throughout training rounds, preventing issues from stale parameters (Wang et al., 2021). With the sigmoid approximation of sign loss and for differentiable $F_k(\mathbf{w})$, our objective $\widetilde{F}(\mathbf{w})$ in (3) is differentiable and can be minimized with gradient descent and its variants. Its gradient is given by:

$$\nabla\widetilde{F}(\mathbf{w}) = \frac{1}{M}\sum_{k=1}^{M}\underbrace{(1-\widetilde{F}_k(\mathbf{w}))\widetilde{F}_k(\mathbf{w})}_{\text{aggregating weight}:=q_k(\mathbf{w})}\ \nabla F_k(\mathbf{w}). \tag{4}$$

Observe that $\nabla\widetilde{F}(\mathbf{w})$ is a **weighted aggregate** of the gradients of the clients' empirical losses, similar in spirit to the gradient $\nabla F(\mathbf{w})$ in standard FL. The key difference is that in MAXFL, the weights $q_k(\mathbf{w}) := (1-\widetilde{F}_k(\mathbf{w}))\widetilde{F}_k(\mathbf{w})$ depend on how much the global model appeals to the clients and are dynamically updated based on the current model $\mathbf{w}$, as we discuss below.

**Behavior of the Aggregation Weights.** For a given $\mathbf{w}$, the aggregation weights $q_k(\mathbf{w})$ depend on the *GM-APPEAL Gap*, $F_k(\mathbf{w}) - \rho_k$ (see Fig. 3). When $F_k(\mathbf{w}) \ll \rho_k$, the global model $\mathbf{w}$ sufficiently meets the client's requirement. Therefore, MAXFL sets $q_k(\mathbf{w}) \approx 0$ to focus on the updates of other clients. Similarly, if $F_k(\mathbf{w}) \gg \rho_k$, MAXFL sets $q_k(\mathbf{w}) \approx 0$. This is because $F_k(\mathbf{w}) \gg \rho_k$ implies that the current model $\mathbf{w}$ is incompatible with the requirement of client $k$ and hence it is better to avoid optimizing for this client at the risk of sacrificing other clients' requirements. MAXFL gives the highest weight to clients for which the global model performs similarly to the clients' requirements since this allows it to increase the GM-APPEAL without sabotaging other clients' requirements.

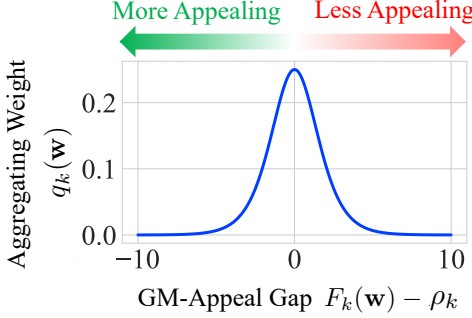

Figure 3: Aggregating weight $q_k(\mathbf{w})$ versus the GM-APPEAL gap defined as $F_k(\mathbf{w}) - \rho_k$ for any client $k \in [M]$.

**A Practical MaxFL Solver.** Directly minimizing the MAXFL objective using gradient descent can be slow to converge and impractical, as it requires all clients to be available for training. Instead, we propose a practical MAXFL algorithm, which uses multiple local updates at each client to speed up convergence as done in standard FL (McMahan et al., 2017) and allow partial client participation. We use the superscript $(t, r)$ to denote the communication round $t$ and the local iteration index $r$. In each round $t$, the server selects a new set of clients $\mathcal{S}^{(t,0)}$ uniformly at random and sends the most recent global model $\mathbf{w}^{(t,0)}$ to the clients in $\mathcal{S}^{(t,0)}$. Clients in $\mathcal{S}^{(t,0)}$ perform $\tau$ local iterations with a learning rate $\eta_l$ to calculate their updates as $\mathbf{w}_k^{(t,r+1)} = \mathbf{w}_k^{(t,r)} - \eta_l \mathbf{g}(\mathbf{w}_k^{(t,r)}, \xi_k^{(t,r)}), \forall\ r \in \{0, ..., \tau-1\}$ where $\mathbf{g}(\mathbf{w}_k^{(t,r)}, \xi_k^{(t,r)}) = \frac{1}{b}\sum_{\xi \in \xi_k^{(t,r)}} \nabla f(\mathbf{w}_k^{(t,r)}, \xi)$ is the stochastic gradient computed using a mini-batch $\xi_k^{(t,r)}$ of size $b$ that is randomly sampled from client $k$'s local dataset $\mathcal{B}_k$. The weight $q_k(\mathbf{w}_k^{(t,0)})$ can be computed at each client by calculating the loss over its training data with $\mathbf{w}_k^{(t,0)}$, which is a simple inference step. Clients in $\mathcal{S}^{(t,0)}$ then send their local updates $\Delta\mathbf{w}_k^{(t,0)} := \mathbf{w}_k^{(t,\tau)} - \mathbf{w}_k^{(t,0)}$ and weights $q_k(\mathbf{w}_k^{(t,0)})$ back to the server, which updates the global model as $\mathbf{w}^{(t+1,0)} = \mathbf{w}^{(t,0)} - \eta_g^{(t,0)}\sum_{k \in \mathcal{S}^{(t,0)}} q_k(\mathbf{w}^{(t,0)})\Delta\mathbf{w}_k^{(t,0)}$ where $\eta_g^{(t,0)} = \frac{\eta_g}{\sum_{k \in \mathcal{S}^{(t,0)}} q_k(\mathbf{w}^{(t,0)}) + \epsilon}$ is the adaptive server learning rate with global learning rate $\eta_g$ and $\epsilon > 0$. We discuss the reasoning for such a learning rate below. The pseudo-code for our MAXFL solver can be found in Algorithm 1.

**Adaptive Server Learning Rate for MaxFL.** With $L_c$ continuous and $L_s$ smooth $F_k(\mathbf{w})$, $\forall k \in [M]$ (see Assumption 3.1), the objective $\widetilde{F}(\mathbf{w})$ is $\widetilde{L}_s$ smooth where $\widetilde{L}_s = \frac{L_s}{M}\sum_{k=1}^{M} q_k(\mathbf{w}) + \frac{L_c}{4}$ (see Appendix C). Hence, the optimal learning rate $\tilde{\eta}$ for the MAXFL is given by, $\tilde{\eta} = 1/\widetilde{L}_s = M\eta/\left(\sum_{k=1}^{M} q_k(\mathbf{w}) + \epsilon\right)$, where $\eta = \frac{1}{L_s}$ is the optimal learning rate for standard FL and $\epsilon = \frac{ML_c}{4L_s} > 0$ is a constant. The denominator of the optimal $\tilde{\eta}$ is proportional to the sum of the aggregation weights $q_k(\mathbf{w})$ and acts as a dynamic normalizing factor. Therefore, we propose using an adaptive global learning rate $\eta_g^{(t,0)} = \eta_g/(\sum_{k \in \mathcal{S}^{(t,0)}} q_k(\mathbf{w}^{(t,0)}) + \epsilon)$ with hyperparameters $\eta_g, \epsilon$.

**Setting $\rho_k$ as $F_k(\widehat{\mathbf{w}}_k)$ for MaxFL.** One intuitive way to set $\rho_k$ for each client is to set it as the training loss value $F_k(\widehat{\mathbf{w}}_k)$ where $\widehat{\mathbf{w}}_k$ is a client local model that is solo-trained with a few warm-up local SGD steps

---

**Algorithm 1** Our Proposed MAXFL Solver

---

1: **Input:** mini-batch size $b$, local iteration steps $\tau$, client requirement $\rho_k$, $k \in [M]$
2: **Output:** Global model $\mathbf{w}^{(T,0)}$
3: **Initialize:** Global model $\mathbf{w}^{(0,0)}$
4: **For** $t = 0, ..., T - 1$ **communication rounds do**:
5:     **Global server do:**
6:       Select $m$ clients for $\mathcal{S}^{(t,0)}$ uniformly at random and send $\mathbf{w}^{(t,0)}$ to clients in $\mathcal{S}^{(t,0)}$
7:     **Clients** $k \in \mathcal{S}^{(t,0)}$ **in parallel do:**
8:       Set $\mathbf{w}_k^{(t,0)} = \mathbf{w}^{(t,0)}$, and calculate $q_k(\mathbf{w}_k^{(t,0)}) = \sigma(F_k(\mathbf{w}_k^{(t,0)}) - \rho_k)$
9:       **For** $r = 0, ..., \tau - 1$ **local iterations do:**
10:         Update $\mathbf{w}_k^{(t,r+1)} \leftarrow \mathbf{w}_k^{(t,r)} - \eta_l \mathbf{g}(\mathbf{w}_k^{(t,r)}, \xi_k^{(t,r)})$
11:       Send $\Delta\mathbf{w}_k^{(t,0)} = \mathbf{w}_k^{(t,0)} - \mathbf{w}_k^{(t,\tau)}$ and aggregation weight $q_k(\mathbf{w}_k^{(t,0)})$ to the server
12:     **Global server do:**
13:       Update global model with $\mathbf{w}^{(t+1,0)} = \mathbf{w}^{(t,0)} - \eta_g^{(t,0)} \sum_{k \in \mathcal{S}^{(t,0)}} q_k(\mathbf{w}^{(t,0)}) \Delta\mathbf{w}_k^{(t,0)}$

---

on its local data. The loss value only needs to be computed once and saved as a constant beforehand at each client. How to train the local model $\widehat{\mathbf{w}}_k$, such as deciding the number of steps of training or whether to add regularization, is dependent on the personal resources and requirements of the clients. Therefore, the quality of the local model (such as whether it has overfitted or underfitted) is not explicitly controlled by MAXFL. Nevertheless, we do provide an ablation study on the number of local SGD steps we take for training the local trained model and provide the results in Appendix D.2 which shows that MAXFL performance is in fact robust to the number of local SGD steps we take and the performance does not vary much on this number. In the essence, $\rho_k$ is a client-dependent parameter that the client can choose, and our proposed MAXFL is a more general framework that can be used for any client-defined $\rho_k$. One might raise concerns that adversarial clients can send arbitrarily small $\rho_k$ in the hope to get high weights, but as Fig. 3 clearly shows this will in fact make the client get smaller weights. Further, we add experiments with byzantine clients in Section 5 in Appendix D.2 and show that MAXFL is robust to these specific attack scenarios.

**Appeal-based Flexible Client Participation.** It may appear that our MAXFL solver requires clients to always participate in FL if selected even when the global model does not appeal to them. However, our algorithm is easily modified to allow clients to participate flexibly during training depending on whether they find the global model appealing or not. For such appeal-based flexible client participation, we assume that clients are available for training if selected only during a few initial training rounds. After these rounds, clients are included in the pool of clients where the server can select the clients from only if they find the global model appealing. We demonstrate this extension of MAXFL with appeal-based flexible client participation in Table 1. The experiment shows that with flexible client participation, retaining a high global model is *even more imperative* for the server to achieve good test accuracy and generalization performance. We also show that even after we allow clients to participate flexibly, MAXFL retains a significantly higher number of clients that find the global model appealing compared to the other baselines.

### 3.1 Convergence Properties of MaxFL

In this section, we show the convergence guarantees of MAXFL in Algorithm 1. Our convergence analysis shows that the gradient norm of our global model goes to zero, and therefore we converge to a stationary point of our objective $\widetilde{F}(\mathbf{w})$. First, we introduce the assumptions and definitions below.

**Assumption 3.1** (Continuity & Smoothness of $F_k(\mathbf{w})$, $\forall\ k$). *The local objective functions $F_1(\mathbf{w})$, ..., $F_M(\mathbf{w})$, are $L_c$-continuous and $L_s$-smooth for any $\mathbf{w}$.*

**Assumption 3.2** (Unbiased Stochastic Gradient with Bounded Variance for $F_k(\mathbf{w})$, $\forall\ k$). *For mini-batch $\xi_k$ uniformly sampled at random from $\mathcal{B}_k$, the resulting stochastic gradient is unbiased, i.e., $\mathbb{E}[\mathbf{g}_k(\mathbf{w}_k, \xi_k)] = \nabla F_k(\mathbf{w}_k)$, and its variance is bounded: $\mathbb{E}[\|\mathbf{g}_k(\mathbf{w}_k, \xi_k) - \nabla F_k(\mathbf{w}_k)\|^2] \leq \sigma_g^2$.*

**Assumption 3.3** (Bounded Dissimilarity of $F(\mathbf{w})$). *There exists $\beta^2 \geq 1$, $\kappa^2 \geq 0$ such that $\frac{1}{M} \sum_{i=1}^M \|\nabla F_i(\mathbf{w})\|^2 \leq \beta^2 \|\frac{1}{M} \sum_{i=1}^M \nabla F_i(\mathbf{w})\|^2 + \kappa^2$ for any $\mathbf{w}$.*

Assumption 3.1-3.3 are standard assumptions used in the optimization literature (Stich, 2019; Karimireddy et al., 2019; Bistritz et al., 2020; Wang et al., 2020), including the $L_c$-continuity assumption (Shalev-Shwartz et al., 2009; Riis et al., 2021). Note that we do not assume anything for our proposed objective function $\widetilde{F}(\mathbf{w})$ and only have assumptions over the standard objective function $F(\mathbf{w})$ to prove the convergence of MAXFL over $\widetilde{F}(\mathbf{w})$ in Theorem 3.1.

**Theorem 3.1** (Convergence to the MAXFL Objective $\widetilde{F}(\mathbf{w})$)**.** *Under Assumption 3.1-3.3, suppose the server uniformly selects $m$ out of $M$ clients without replacement in each round of Algorithm 1. With $\eta_l = \frac{1}{\sqrt{T}\tau}$, $\eta_g = \sqrt{\tau m}$, for total communication rounds $T$ of the MAXFL solver in Algorithm 1 we have:*

$$\min_{t\in[T]} \mathbb{E}\left[\left\|\nabla\widetilde{F}(\mathbf{w}^{(t,0)})\right\|^2\right] \leq \mathcal{O}\left(\frac{\sigma_g^2}{\sqrt{m\tau T}}\right) + \mathcal{O}\left(\frac{\sigma_g^2}{T\tau}\right) + \mathcal{O}\left(\frac{\sqrt{\tau}}{\sqrt{Tm}}\right) + \mathcal{O}\left(\frac{\kappa^2+\beta^2}{T}\right) \tag{5}$$

*where $\mathcal{O}$ subsumes all constants (including $L_s$ and $L_c$).*

Theorem 3.1 shows that with a sufficiently large number of communication rounds $T$ we reach a stationary point of our objective function $\widetilde{F}(\mathbf{w})$. The proof is deferred to Appendix C where we also show a version of this theorem that contains the learning rates $\eta_g$ and $\eta_l$ with the constants.

## 4 Related Work

To the best of our knowledge, the notion of GM-APPEAL and the proposal to maximize it while considering flexible client participation have not appeared before in the previous literature. Previous works have focused on the notion of satisfying clients' personal requirements from a game-theoretic lens or designing strategies specifically to prevent client dropout, including the use of personalization, which have their limitations, as we discuss below.

### 4.1 Incentivizing Clients and Preventing drop-out

A recent line of work in game theory models FL as a coalition of self-interested agents and studies how clients can optimally satisfy their individual incentives defined differently from our goal. Instead of training a single global model, Donahue & Kleinberg (2021a;b) consider the problem where each client tries to find the best possible coalition of clients to federate with to minimize its own error. Blum et al. (2021) consider an orthogonal setting where each client aims to satisfy its constraint of low expected error while simultaneously trying to minimize the number of samples it contributes to FL. While these works establish useful insights for simple linear tasks, it is difficult to extend these to practical non-convex machine learning tasks. In contrast to these works, in MAXFL we aim to directly maximize the *number* of satisfied clients using a global model. Concurrent work (Huang et al., 2023) has evaluated the diverse data contributions in FL where the tension between the server and the clients is modeled as a utility-cost function to analyze the optimal behavior of clients in FL and propose a mechanisam to incentivize agents to give contribution to FL using the Stackelberg game to maximize the total utility. Another concurrent similar line of work (Dorner et al., 2023) has investigated how to incentivize "honesty" across clients in FL where honesty implies clients sending local updates that are true to their local data and non-malicious. This perspective alleviates some of the analysis complexities occurring in game-theoretic formulations and allows us to consider general non-convex objective functions.

A separate line of work looks at how to prevent and deal with client drop-out in FL. Wang & Xu (2022) introduce a notion of 'friendship' among clients and proposes to use friends' local update as a substitute for the update of dropped-out clients. Gu et al. (2021) propose to use previous updates of dropped-out clients as a substitute for their current updates. Both algorithms are stateful. Another line of work (Han et al., 2022; Kang et al., 2019; Zhang et al., 2021) aims to incentivize clients to contribute resources for FL and promote long-term participation by providing monetary compensation for their contributions, determined using game-theoretic tools. These techniques are orthogonal to MAXFL's formulation and can be combined if needed to further incentivize clients.

### 4.2 Personalized and Fair Federated Learning

Personalized federated learning (PFL) methods aim to increase performance by training multiple related models across the network (e.g., Smith et al., 2017). In contrast to PFL, MAXFL focuses on the challenging goal of

training a *single* global model that can maximize the number of clients for which the global model outperforms their local model. Unlike PFL which may require additional training on new clients for personalization, MaxFL's global model can be used by new clients without additional training (see Table 2). Also, MaxFL is stateless, in that clients do not carry varying local parameters throughout training rounds as in many popular personalized FL methods (Smith et al., 2017; Dinh et al., 2020; Fallah et al., 2020; Li et al., 2021), preventing parameter staleness problems which can be exacerbated by partial client participation (Wang et al., 2021). Furthermore, MaxFL is orthogonal to and can be combined with PFL methods. We demonstrate this in Table 3, where we show results for MaxFL jointly used with personalization via fine-tuning (Jiang et al., 2019). We compare MaxFL +Fine-tuning with another well known PFL method PerFedAvg (Fallah et al., 2020) and show that MaxFL appeals to a significantly higher number of clients than the baseline.

Finally, another related area is fair FL, where a common goal is to train a global model whose accuracy has less variance across the client population than standard FedAvg (Li et al., 2019; Mohri et al., 2019). A side benefit of these methods is that they can improve global model appeal for the worst performing clients. However, the downside is that the performance of the global model may be degraded for the best performing clients, thus making it unappealing for them to participate. We show in Appendix D.2 that fair FL methods are indeed not effective in increasing GM-Appeal.

## 5 Experiments

In this section we evaluate MaxFL for a number of different datasets while comparing with a wide range of baselines to show that maximizing GM-Appeal, i.e., training a global model that can appeal to a *larger number of clients*, provide many benefits for FL including: i) the server gaining more participating clients to select clients from for training a better global model for the seen clients, ii) the global model having a higher chance to have a good performance on unseen clients, and iii) clients gaining better performance with the global model when they combine MaxFL with local fine-tuning.

**Datasets and Model.** We evaluate MaxFL in three different settings: image classification for non-iid partitioned (i) FMNIST (Xiao et al., 2017), (ii) EMNIST with 62 labels (Cohen et al., 2017), and (iii) sentiment analysis for (iv) Sent140 (Go et al., 2009)with a MLP. For FMNIST, EMNIST, and Sent140 dataset, we consider 100, 500, and 308 clients in total that are used for training where we select 5 and 10 clients uniformly at random per round for FMNIST and EMNIST, Sent140 respectively. These clients are active at some point in training the global model and we call them **'seen clients'**. We also sample the **'unseen clients'** from the same distribution from which we generate the seen clients, with 619 clients for Sent140, 100 clients for FMNIST, and 500 for EMNIST. These unseen clients represent new incoming clients that have not been seen before during the training rounds of FL to evaluate the generalization performance at inference. Further details of the experimental settings are deferred to Appendix D.1.

**Baselines.** We compare MaxFL with numerous well-known FL algorithms such as standard FedAvg (McMahan et al., 2017); FedProx (Sahu et al., 2020) which aims to tackle data heterogeneity; SCAFFOLD (Karimireddy et al., 2019) which aims for variance-reduction; PerFedAvg (Fallah et al., 2020), pFedme (Dinh et al., 2020) which facilitates personalization; MW-Fed (Blum et al., 2021) which incentivizes client participation; and qFFL which facilitates fairness (Li et al., 2019). For all algorithms, we set $\rho_k$ to be the same, i.e., $\rho_k = F_k(\widehat{\mathbf{w}}_k)$, where $\widehat{\mathbf{w}}_k$ is obtained by running a few warm-up local SGD steps on client $k$'s data as outlined in Section 3 to ensure a fair comparison across baselines. We perform grid search for hyperparameter tuning for all baselines and choose the best performing ones.

**Evaluation Metrics: GM-Appeal, Average Test Accuracy, and Preferred-model Test Accuracy.** We evaluate MaxFL and other methods with three key metrics: 1) GM-Appeal, defined in (1), 2) average test accuracy (avg. test acc.) across clients, and a new metric that we propose called 3) preferred-model test accuracy. Preferred-model test accuracy is the average of the clients' test accuracies computed on either the global model $\mathbf{w}$ or their solo-trained local model $\widehat{\mathbf{w}}_k$, whichever one satisfies the client's requirement. We belive that average test accuracy is a more server-oriented metric as it assumes that clients will use the global model by default. On the other hand, preferred-model test accuracy is a more client-centric metric that allows clients to select the model which works best, thereby better reflecting their actual satisfaction.

|  | Seen Clients | | | | Unseen Clients | | | |
|  | FMNIST | | EMNIST | | FMNIST | | EMNIST | |
|  | Test Acc. | GM-Appeal | Test Acc. | GM-Appeal | Test Acc. | GM-Appeal | Test Acc. | GM-Appeal |
|---|---|---|---|---|---|---|---|---|
| FedAvg | $43.70(\pm0.02)$ | $0.04(\pm0.0)$ | $35.15(\pm0.51)$ | $0.02(\pm0.01)$ | $43.14(\pm0.23)$ | $0.07(\pm0.01)$ | $37.14(\pm0.10)$ | $0.06(\pm0.0)$ |
| FedProx | $44.59(\pm1.94)$ | $0.05(\pm0.01)$ | $34.06(\pm1.21)$ | $0.004(\pm0.0)$ | $43.80(\pm1.67)$ | $0.07(\pm0.01)$ | $36.82(\pm0.22)$ | $0.008(\pm0.0)$ |
| Scaffold | $39.90(\pm0.59)$ | $0.0(\pm0.0)$ | $34.78(\pm2.05)$ | $0.0(\pm0.0)$ | $39.24(\pm0.68)$ | $0.01(\pm0.0)$ | $34.19(\pm1.25)$ | $0.004(\pm0.0)$ |
| PerFedAvg | $46.62(\pm1.0)$ | $0.05(\pm0.0)$ | $34.78(\pm1.05)$ | $0.003(\pm0.0)$ | $46.00(\pm0.87)$ | $0.07(\pm0.0)$ | $36.92(\pm0.51)$ | $0.008(\pm0.0)$ |
| pFedme | $31.06(\pm2.06)$ | $0.0(\pm0.0)$ | $9.78(\pm2.13)$ | $0.0(\pm0.0)$ | $20.11(\pm3.4)$ | $0.0(\pm0.0)$ | $7.05(\pm1.03)$ | $0.0(\pm0.0)$ |
| qFFL | $29.92(\pm3.13)$ | $0.0(\pm0.0)$ | $15.95(\pm3.02)$ | $0.0(\pm0.0)$ | $19.63(\pm2.17)$ | $0.0(\pm0.0)$ | $5.41(\pm0.52)$ | $0.0(\pm0.0)$ |
| MW-Fed | $44.41(\pm2.38)$ | $0.04(\pm0.0)$ | $30.44(\pm3.07)$ | $0.01(\pm0.0)$ | $43.46(\pm2.15)$ | $0.06(\pm0.0)$ | $36.54(\pm0.40)$ | $0.01(\pm0.0)$ |
| MaxFL | $\mathbf{70.86}(\pm2.18)$ | $\mathbf{0.37}(\pm0.05)$ | $\mathbf{57.34}(\pm1.41)$ | $\mathbf{0.25}(\pm0.03)$ | $\mathbf{74.53}(\pm0.50)$ | $\mathbf{0.39}(\pm0.07)$ | $\mathbf{55.62}(\pm0.86)$ | $\mathbf{0.31}(\pm0.03)$ |

Table 1: Avg. test accuracy and GM-Appeal where we train for 200 communication rounds. At the 10th communication round, we let clients flexibly opt-out or opt-in depending on whether the global model has met their requirements. We report the final avg. test accuracy and GM-Appeal at the 200th comm. round.

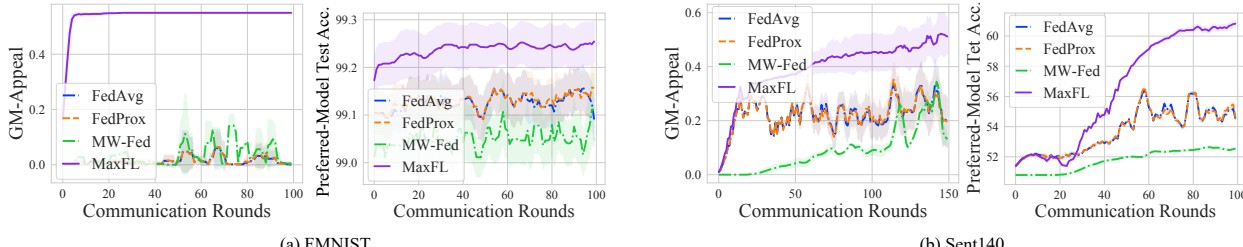

(a) FMNIST                 (b) Sent140

Figure 4: GM-Appeal and preferred-model test accuracy for the seen clients are significantly higher for MaxFL. Therefore, clients can also benefit from choosing either the local or global model for best performance, while the server also gains a large number of clients to select from.

Note that the preferred-model test accuracy is a complementary novel metric we propose to gauge the satisfaction of the clients when they can choose between the global and local models. For instance, if we have a high preferred-model test accuracy but a low GM-Appeal where all clients are not interested in the global model, then the primary objective of MaxFL has not been achieved. However, in our results we show that in fact MaxFL has the complementary effect of not only being able to maximize the GM-Appeal, but also have higher preferred-model test accuracy. This is because MaxFL is able to find the global model that can perform better than a prefixed threshold for as many clients as possible, and therefore compared to the other global models trained from different baselines, the test accuracy for those clients who chose the global model is higher.

## 5.1 Experiment Results

**Average Test Accuracy of Seen Clients & Unseen Clients.** We first show that we improve the GM-Appeal and thus the average test accuracy performance for the 'seen clients' used during the training of the global model. In Table 1, we show the average test accuracy across clients where we let clients flexibly join or drop-out depending on whether the global model is appealing after 5% of communication rounds of mandatory participation. We show that MaxFL achieves the highest GM-Appeal than other baselines for both FMNIST and EMNIST by 0.32-0.39 and 0.23-0.31 improvement, respectively. Since MaxFL is able to retain a larger pool of clients due to having a higher GM-Appeal, it therefore trains from selecting from a more larger client pool, leading to the highest average test accuracy compared to the baselines by 22-40% and 18-50% improvement respectively for the seen and unseen clients. Since the other baselines do not consider the notion of GM-Appeal entirely, it fails in preventing client dropouts leading to poor performance. Note

that we do not use any of the 'unseen clients' during training and only calculate the GM-APPEAL and test accuracy via inference with the global model trained with the 'seen clients'.

**Preferred-model Test Accuracy: Clients' Perspective.** Recall that a high preferred-model test accuracy implies that the client has a higher chance of satisfying its requirement by choosing between the global or solo-trained local model, whichever performs better. In Fig. 4 we show that as the GM-APPEAL increases across the communication round, preferred-model test accuracy also increases. MAXFL achieves the highest final GM-APPEAL and preferred-model test accuracy indicating that it provides a win-win case for both the server and clients, since the clients have the highest accuracy by choosing the better model between the global model $\mathbf{w}$ and the local model $\widehat{\mathbf{w}}_k$,

|  | GM-APPEAL | | Preferred-Model Test Acc. | |
|---|---|---|---|---|
|  | FMNIST | Sent140 | FMNIST | Sent140 |
| FedAvg | $0.08(\pm0.01)$ | $0.37(\pm0.07)$ | $98.53(\pm0.13)$ | $57.05(\pm1.44)$ |
| FedProx | $0.07(\pm0.01)$ | $0.37(\pm0.07)$ | $98.43(\pm0.21)$ | $57.07(\pm1.42)$ |
| Scaffold | $0.02(\pm0.01)$ | $0.03(\pm0.05)$ | $98.26(\pm0.20)$ | $51.59(\pm0.11)$ |
| MW-Fed | $0.05(\pm0.04)$ | $0.17(\pm0.03)$ | $98.32(\pm0.13)$ | $55.57(\pm1.28)$ |
| MAXFL | $\mathbf{0.55}(\pm0.0)$ | $\mathbf{0.43}(\pm0.05)$ | $\mathbf{98.83}(\pm0.06)$ | $\mathbf{57.16}(\pm1.35)$ |

Table 2: GM-APPEAL and preferred-model test accuracy of the final global models for the unseen clients. MAXFL improves the GM-APPEAL by at least 47% for FMNIST, and 6% for Sent140 and achieves the same or higher preferred-model test accuracy.

and the server has the highest fraction of participating clients. Similarly, in Table 2, MAXFL achieves the highest GM-APPEAL and preferred-model test accuracy. Although the preferred-model test accuracy improvement compared to the other baselines may appear small, showing that MAXFL is able to maintain a high preferred-model test accuracy while also achieving a high GM-APPEAL implies that it does not sabotage the benefit of clients while also bringing the server more clients to select from.

**Local Tuning for Personalization.** Personalized FL methods can be used to fine-tune the global model at each client before comparing it with the client's locally trained model. MAXFL can be combined with these methods by simply allowing clients to perform some fine-tuning iterations before computing the aggregation weights in Step 7 of Algorithm 1. Both for clients that are active during training and unseen test clients, we show in Table 3 that MAXFL increases the GM-APPEAL by at least 10% compared to all baselines. For FMNIST and Sent140, the improvement in GM-APPEAL over other methods is up to 27%, 28% respectively for active clients and 17%, 4% respectively for unseen clients.

|  | Seen Clients | | Unseen Clients | |
|---|---|---|---|---|
|  | FMNIST | Sent140 | FMNIST | Sent140 |
| FedAvg | $0.38(\pm0.06)$ | $0.25(\pm0.09)$ | $0.39(\pm0.06)$ | $0.42(\pm0.06)$ |
| FedProx | $0.40(\pm0.07)$ | $0.26(\pm0.09)$ | $0.41(\pm0.07)$ | $0.43(\pm0.12)$ |
| Scaffold | $0.02(\pm0.02)$ | $0.16(\pm0.22)$ | $0.03(\pm0.02)$ | $0.07(\pm0.01)$ |
| PerFedAvg | $0.45(\pm0.05)$ | $0.24(\pm0.10)$ | $0.46(\pm0.06)$ | $0.47(\pm0.06)$ |
| MW-Fed | $0.28(\pm0.07)$ | $0.08(\pm0.01)$ | $0.39(\pm0.04)$ | $0.20(\pm0.01)$ |
| MAXFL | $\mathbf{0.55}(\pm0.01)$ | $\mathbf{0.36}(\pm0.05)$ | $\mathbf{0.56}(\pm0.01)$ | $\mathbf{0.55}(\pm0.01)$ |

Table 3: GM-APPEAL of locally-tuned models with 5 local steps from the final global models for seen clients and unseen clients. Both for clients that are active during training and unseen test clients, MAXFL increases the fraction of clients that find the global model appealing by at least 10% as compared to all baselines.

## 6 Limitations and Concluding Remarks

In this work we explore the notion of global model appeal by proposing to train a global model that maximizes the number of clients whose local requirements are satisfied. We show that when participating clients drop out or clients do not join due to a low global model appeal, the test accuracy for the current clients and generalization performance to the new unseen clients can suffer significantly. Through extensive experiments as well as theoretical insights and guarantees, we show that MAXFL can help to retain clients for training and thus achieve a high average test accuracy across both participating clients and new incoming clients.

We note that our proposed metric GM-APPEAL and MAXFL objective have some limitations. For instance, it is possible to train a global model that sacrifices the performance of a few clients to maximize GM-APPEAL, potentially reducing fairness. However, we expect that MAXFL could potentially be altered by modulating the $\rho_k$ value to cover such limitations such as setting $\rho_k$ to a constant that retains fairness. Another limitation

in MaxFL is that it does not currently consider specific incentive mechanisms for various settings such as what cost we can set for the new incoming clients that wants to use the global model at inference without participating in training. Without setting this cost, one may raise the concern of free-rider problems. Nevertheless, our work presents a first step towards maximizing the set of clients that are interested in the global model; how to design the cost of using a global model would be an interesting, orthogonal direction of future work. As a similar notion of global model appeal has not been thoroughly examined previously, we hope our work can open up new research directions in understanding the role played by the server to prevent client dropout and recruit new clients by finding a global model that serves as many clients as possible.

**Broader Impact Statement**

Our work proposes a new objective to maximize global model appeal in federated learning, which can incentivize more clients to participate and prevent clients from dropping out, and improve generalization performance on unseen clients. Despite these benefits, it is worth acknowledging the potential negative effects of the proposed objective and algorithm in terms of other metrics, such as unfairness (e.g., increased performance gap) between different sub-populations. Here, we note that (1) Our MaxFL framework is general in the sense that it could be adjusted to trade off fairness/utility by setting the local requirement parameters $\rho_k$ appropriately for specific sub-populations, arriving at a fairer model that reduces the number of clients suffering from inferior performance; and (2) Our results demonstrate that MaxFL is in fact more robust to specific Byzantine clients that adversarially send high GM-Appeal gap than competitors due to its dynamic reweighting scheme (see Table 4). In general, however, it remains critical to carefully consider trade-offs between issues such as model appeal, fairness, and robustness for the application at hand. The goal of this work is to explore the implications of MaxFL both theoretically and empirically so that we can understand various benefits and limitations of GM-Appeal in different scenarios. We hope practitioners and researchers can thus appropriately adjust the framework and/or combine it with other learning schemes depending on the application of interest, weighing potential benefits of the approach relative to existing methods in terms of achieving higher accuracy, broader participation, and increased fairness or robustness.

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

# A  Further Results and Details for Our Motivating Toy Example

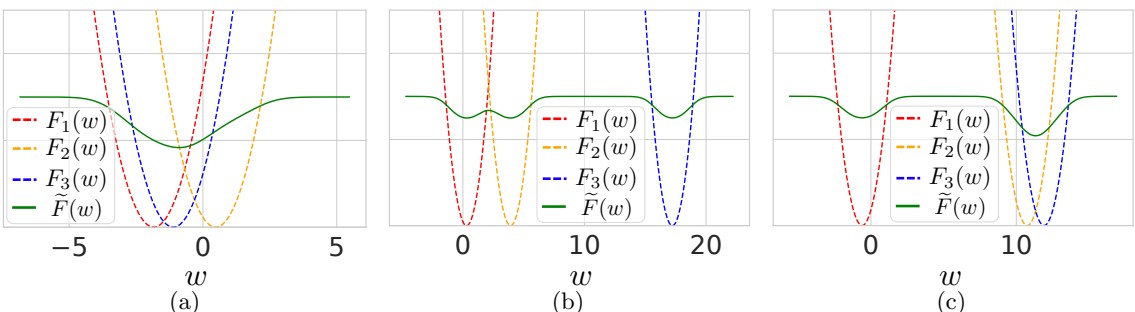

Figure 5: Results for the three client mean estimation; (a): case 1 when the true mean across clients are close to amongst each other where MAXFL's optimal solution is identical to that of FedAvg; (b): case 2 when the true mean across clients are all different from each other where MAXFL's optimal solution ensures that at least one of the clients will be satisfied with MAXFL's global model (unlike FedAvg); (c) case 3 when two clients' true means are close to each other while the other client has a different mean. MAXFL in this case, is able to ensure that the two clients satisfied while FedAvg is not able to make any client satisfied.

**Mean Estimation with 3 Clients with MaxFL.** We further examine the property of MAXFL to satisfy clients with a 3 clients toy example which is an extension from what we have shown for 2 clients. Reusing the notation from the 2 client example, where $\theta_i$ is the true mean at client $i$ and $\hat{\theta}_i \sim \mathcal{N}(\theta_i, 1)$ is the empirical mean of a client, our analysis can be divided into the following cases for the 3 client example (also depicted in Fig. 5):

- Case 1: $\theta_1 \approx \theta_2 \approx \theta_3$: This case captures the setting where the data at the clients is almost i.i.d. In this case, it makes sense for clients to collaborate together and therefore MAXFL's optimal solution will be the average of local empirical means (same as FedAvg).

- Case 2: $\theta_1 \neq \theta_2 \neq \theta_3$: This case captures the setting where the data at clients is completely disparate. In this case, none of the clients benefit from collaborating and therefore MAXFL's optimal solution will be the local model of one of the clients. This ensures at least one of the clients will still be satisfied with the MAXFL global model unlike FedAvg.

- Case 3: $\theta_1 \approx \theta_2 \neq \theta_3$: The most interesting case happens when data at two of the clients is similar but the data at the third client is different. Without loss of generality we assume that data at clients 1 and 2 is similar and client 3 is different. In this case, although client 1 and 2 benefit from federating, FedAvg is unable to leverage that due to the heterogeneity at client 3. MAXFL, on the other hand, will set the optimal solution to be the average of the local models of just client 1 and client 2. This ensures clients 1 and 2 are satisfied with the global model, thus maximizing the GM-APPEAL.

**Simulation Details for Fig. 2** For the mean estimation simulation for Fig. 2(a), we set the true means for the two clients as $\theta_1 = 0$, $\theta_2 = 2\gamma_G$ where $\gamma_G \in [0, \sqrt{20}]$. The simulation was perfomed using NumPy (Harris et al., 2020) and SciPy (Virtanen et al., 2020). The empirical means $\hat{\theta}_1$ and $\hat{\theta}_2$ are sampled from the distribution $\mathcal{N}(\theta_1, 1)$ and $\mathcal{N}(\theta_2, 1)$ respectively where the number of samples are assumed to be identical for simplicity. For local training we assume clients set their local models as their local empirical means which is analogous to clients performing a large number of local SGD steps to obtain the local minima of their empirical loss. For the global objective (standard FL, MAXFL (ReLU), MAXFL) a local minima is found using the `scipy.optimize` function in the SciPy package. For each $\gamma_G^2 \in [0, \sqrt{20}]$, the average GM-Appeal is calculated over 10000 runs for each global objective.

# B    Proof for Theoretical Analysis in Section 2.2

We additionally define the following quantities

$$\gamma^2 := \frac{\nu^2}{N}; \quad \gamma_G^2 = \left(\frac{\theta_2 - \theta_1}{2}\right)^2; \tag{6}$$

Note that the distribution of the empirical means itself follows a normal distribution following the linear additivity of independent normal random variables.

$$\widehat{\theta}_1 \sim \mathcal{N}(\theta_1, \gamma^2); \quad \widehat{\theta}_2 \sim \mathcal{N}(\theta_2, \gamma^2) \tag{7}$$

**Lemma A.1** *The expected GM-Appeal of the standard FL model is upper bounded by $2\exp\left(-\frac{\gamma_G^2}{5\gamma^2}\right)$, where the expectation is taken over the randomness in the local datasets $\mathcal{B}_1, \mathcal{B}_2$.*

**Proof.**

The standard FL model is given by, $w = \frac{\widehat{\theta}_1 + \widehat{\theta}_2}{2}$, and therefore the expected GM-Appeal is,

$$\mathbb{E}\left[\frac{\mathbb{I}\{(w - \theta_1)^2 < (\widehat{\theta}_1 - \theta_1)^2\} + \mathbb{I}\{(w - \theta_2)^2 < (\widehat{\theta}_2 - \theta_2)^2\}}{2}\right] \tag{8}$$

$$= \frac{1}{2}\left[\underbrace{\mathbb{P}\left((w - \theta_1)^2 < (\widehat{\theta}_1 - \theta_1)^2\right)}_{T_1} + \underbrace{\mathbb{P}\left((w - \theta_2)^2 < (\widehat{\theta}_2 - \theta_2)^2\right)}_{T_2}\right] \tag{9}$$

Next we bound $T_1$ and $T_2$.

$$T_1 = \mathbb{P}\left((w - \theta_1)^2 < (\widehat{\theta}_1 - \theta_1)^2\right) \tag{10}$$

$$= \mathbb{P}\left(\left(\frac{\widehat{\theta}_1 + \widehat{\theta}_2}{2} - \theta_1\right)^2 < (\widehat{\theta}_1 - \theta_1)^2\right) \tag{11}$$

$$= \mathbb{P}\left(\left(\frac{\widehat{\theta}_2 - \widehat{\theta}_1}{2}\right)^2 + 2\left(\frac{\widehat{\theta}_2 - \widehat{\theta}_1}{2}\right)(\widehat{\theta}_1 - \theta_1) < 0\right) \tag{12}$$

$$= \mathbb{P}\left(\left\{\left(\frac{\widehat{\theta}_2 - \widehat{\theta}_1}{2}\right)^2 + 2\left(\frac{\widehat{\theta}_2 - \widehat{\theta}_1}{2}\right)(\widehat{\theta}_1 - \theta_1) < 0\right\} \cap \left\{\widehat{\theta}_2 > \widehat{\theta}_1\right\}\right)$$

$$+ \mathbb{P}\left(\left\{\left(\frac{\widehat{\theta}_2 - \widehat{\theta}_1}{2}\right)^2 + 2\left(\frac{\widehat{\theta}_2 - \widehat{\theta}_1}{2}\right)(\widehat{\theta}_1 - \theta_1) < 0\right\} \cap \left\{\widehat{\theta}_2 \leq \widehat{\theta}_1\right\}\right) \tag{13}$$

$$= \mathbb{P}\left(\left\{\left(\frac{\widehat{\theta}_2 - \widehat{\theta}_1}{2}\right) + 2(\widehat{\theta}_1 - \theta_1) < 0\right\} \cap \left\{\widehat{\theta}_2 > \widehat{\theta}_1\right\}\right)$$

$$+ \mathbb{P}\left(\left\{\left(\frac{\widehat{\theta}_2 - \widehat{\theta}_1}{2}\right)^2 + 2\left(\frac{\widehat{\theta}_2 - \widehat{\theta}_1}{2}\right)(\widehat{\theta}_1 - \theta_1) < 0\right\} \cap \left\{\widehat{\theta}_2 \leq \widehat{\theta}_1\right\}\right) \tag{14}$$

$$\leq \mathbb{P}\left(\left(\frac{\widehat{\theta}_2 - \widehat{\theta}_1}{2}\right) + 2(\widehat{\theta}_1 - \theta_1) < 0\right) + \mathbb{P}\left(\widehat{\theta}_2 - \widehat{\theta}_1 \leq 0\right) \tag{15}$$

$$= \mathbb{P}\left(Z_1 < 0\right) + \mathbb{P}\left(Z_2 \leq 0\right) \quad \text{where } Z_1 \sim \mathcal{N}\left(\gamma_G, \frac{5}{2}\gamma^2\right), Z_2 \sim \mathcal{N}\left(2\gamma_G, 2\gamma^2\right) \tag{16}$$

$$\leq \exp\left(-\frac{\gamma_G^2}{5\gamma^2}\right) + \exp\left(-\frac{\gamma_G^2}{\gamma^2}\right) \tag{17}$$

$$\leq 2\exp\left(-\frac{\gamma_G^2}{5\gamma^2}\right) \tag{18}$$

where (13) uses $\mathbb{P}(A) = \mathbb{P}(A \cap B) + \mathbb{P}\left(A \cap B^{\complement}\right)$, (15) uses $\mathbb{P}(A \cap B) \leq \mathbb{P}(A)$, (16) uses (7) and linear additivity of independent normal random variables, (17) uses a Chernoff bound.

We can similarly bound $T_2$ to get $T_2 \leq 2\exp\left(-\frac{\gamma_G^2}{5\gamma^2}\right)$. Thus the expected GM-Appeal of the standard FL model is upper bounded by $2\exp\left(-\frac{\gamma_G^2}{5\gamma^2}\right)$.

**Lemma A.2** *Let $h$ be any function that is convex, twice differentiable, and strictly increasing in $[0, \infty)$. Then our relaxed objective is strictly convex and has a unique minimizer at $w^* = \left(\frac{\widehat{\theta}_1 + \widehat{\theta}_2}{2}\right)$.*

**Proof.**

Let us denote our relaxed objective by $v(w)$. Then $v(w)$ can be written as,

$$v(w) = \frac{1}{2}\left[h\left(F_1(w) - F(\widehat{w}_1)\right) + h\left(F_2(w) - F(\widehat{w}_2)\right)\right] = \underbrace{\frac{1}{2}h\left((w - \widehat{\theta}_1)^2\right)}_{v_1(w)} + \underbrace{\frac{1}{2}h\left((w - \widehat{\theta}_2)^2\right)}_{v_2(w)} \tag{19}$$

$$\tag{20}$$

We first prove that $v_1(w)$ is strictly convex. Let $\lambda \in (0,1)$ and $(w_1, w_2)$ be any pair of points in $\mathbb{R}^2$ such that $w_1 \neq w_2$. We have,

$$v_1(\lambda w_1 + (1-\lambda)w_2) = \frac{1}{2}h\left((\lambda(w_1 - \widehat{\theta}_1) + (1-\lambda)(w_2 - \widehat{\theta}_1))^2\right) \tag{21}$$

$$< \frac{1}{2}h\left(\lambda(w_1 - \widehat{\theta}_1)^2 + (1-\lambda)(w_2 - \widehat{\theta}_1)^2\right) \tag{22}$$

$$\leq \frac{\lambda}{2}h\left((w_1 - \widehat{\theta}_1)^2\right) + \frac{1-\lambda}{2}h\left((w_2 - \widehat{\theta}_1)^2\right) \tag{23}$$

$$= \lambda v_1(w_1) + (1-\lambda)v_1(w_2) \tag{24}$$

where (22) follows from the strict convexity of $f(w) = w^2$ and the fact that $h(w)$ is strictly increasing in the range $[0, \infty)$, (23) follows from the convexity of $h(w)$.

This completes the proof that $v_1(w)$ is strictly convex. We can similarly prove that $v_2(w)$ is stricly convex and hence $v(w)$ is strictly convex since summation of strictly convex functions is strictly convex.

Also note that,

$$\nabla v(w) = \nabla h\left((w - \widehat{\theta}_1)^2\right)(w - \widehat{\theta}_1) + \nabla h\left((w - \widehat{\theta}_2)^2\right)(w - \widehat{\theta}_2) \tag{25}$$

It is easy to see that $\nabla v(w) = 0$ at $w = \left(\frac{\widehat{\theta}_1 + \widehat{\theta}_2}{2}\right)$. Since $v(w)$ is strictly convex this implies that $w^* = \left(\frac{\widehat{\theta}_1 + \widehat{\theta}_2}{2}\right)$ will be a unique global minimizer. This completes the proof.

**Proof of Theorem A.1**

Before stating the proof of Theorem A.1 we first state some intermediate results that will be used in the proof.

The MAXFL objective can be written as,

$$v(w) = \frac{1}{2}\sigma\left((w - \widehat{\theta}_1)^2\right) + \frac{1}{2}\sigma\left((w - \widehat{\theta}_2)^2\right) \tag{26}$$

where $\sigma(w) = 1/(1 + \exp(-w))$.

We additionally define the following quantities,

$$i := \operatorname{argmin}\left\{\widehat{\theta}_1, \widehat{\theta}_2\right\}; \quad j := \operatorname{argmax}\left\{\widehat{\theta}_1, \widehat{\theta}_2\right\}; \quad \widehat{\gamma}_G := \frac{\widehat{\theta}_j - \widehat{\theta}_i}{2} \tag{27}$$

Let $q(w) = \sigma(w)(1 - \sigma(w))$. The gradient of $v(w)$ is given as,

$$\nabla v(w) = q\left((w - \widehat{\theta}_1)^2\right)(w - \widehat{\theta}_1) + q\left((w - \widehat{\theta}_2)^2\right)(w - \widehat{\theta}_2) \tag{28}$$

**Lemma A.3**  *For $\widehat{\gamma}_G > 2$, $w = \left(\frac{\widehat{\theta}_1 + \widehat{\theta}_2}{2}\right)$ will be a local maxima of the MAXFL objective.*

It is easy to see that $w = \left(\frac{\widehat{\theta}_1 + \widehat{\theta}_2}{2}\right)$ will always be a stationary point of $\nabla v(w)$. Our goal is to determine whether it will be a local minima or a local maxima. To do so, we calculate the hessian of $v(w)$ as follows. Let $f(w) = 2\sigma(w)(1 - \sigma(w))(1 - 2\sigma(w))$. Then,

$$\nabla^2 v(w) = \underbrace{f\left((w - \widehat{\theta}_1)^2\right)(w - \widehat{\theta}_1)^2 + q\left((w - \widehat{\theta}_1)^2\right)}_{h_1(w)} + \underbrace{f\left((w - \widehat{\theta}_2)^2\right)(w - \widehat{\theta}_2)^2 + q\left((w - \widehat{\theta}_2)^2\right)}_{h_2(w)} \tag{29}$$

Note that $h_1(w) = h_2(w)$ for $w = \left(\frac{\widehat{\theta}_1 + \widehat{\theta}_2}{2}\right)$. Hence it suffices to focus on the condition for which $h_1(w) < 0$ at $w = \left(\frac{\widehat{\theta}_1 + \widehat{\theta}_2}{2}\right)$. We have,

$$h_1\left((\widehat{\theta}_1 + \widehat{\theta}_2)/2\right) = f(\widehat{\gamma}_G^2)\widehat{\gamma}_G^2 + q(\widehat{\gamma}_G^2) \tag{30}$$

$$= q(\widehat{\gamma}_G^2)(2(1 - 2\sigma(\widehat{\gamma}_G^2))\widehat{\gamma}_G^2 + 1) \tag{31}$$

$$< 0 \quad \text{for } \widehat{\gamma}_G \geq 1.022 \tag{32}$$

where the last inequality follows from the fact that $q(w) > 0$ for all $w \in \mathbb{R}$ and $2(1 - 2\sigma(w^2))w^2 + 1 < 0$ for $w \geq 1.022$. Thus for $\widehat{\gamma}_G > 2$, $w = \left(\frac{\widehat{\theta}_1 + \widehat{\theta}_2}{2}\right)$ will be a local maxima of the MAXFL objective.

**Lemma A.4**  *For $\widehat{\gamma}_G > 0$, any local minima of $v(w)$ lies in the range $(\widehat{\theta}_i, \widehat{\theta}_i + 2] \cup [\widehat{\theta}_j - 2, \widehat{\theta}_j)$.*

Firstly note that since $\widehat{\gamma}_G > 0$ we have $\widehat{\theta}_j > \widehat{\theta}_i$. Secondly note that since $q(w) > 0$ for all $w \in \mathbb{R}$, $\nabla v(w) < 0$ for all $w \leq \widehat{\theta}_i$ and $\nabla v(w) > 0$ for all $w \geq \widehat{\theta}_j$. Therefore any root of the function $\nabla v(w)$ must lie in the range $(\widehat{\theta}_i, \widehat{\theta}_j)$.

**Case 1:** $0 < \widehat{\gamma}_G \leq 2$.

In this case, the lemma is trivially satisified since $(\widehat{\theta}_i, \widehat{\theta}_j) \subset \left\{ (\widehat{\theta}_i, \widehat{\theta}_i + 2] \cup [\widehat{\theta}_j - 2, \widehat{\theta}_j) \right\}$.

**Case 2:** $\widehat{\gamma}_G > 2$.

Let $x = w - \widehat{\theta}_i$ and $g(x) = q(x^2)x$. We can write $\nabla v(w)$ as, $\nabla v(\widehat{\theta}_i + x) = g(x) - g(2\widehat{\gamma}_G - x)$. It can be seen that for $x > 2$, $g(x)$ is a decreasing function. For $x \in (2, \widehat{\gamma}_G)$ we have $x > 2\widehat{\gamma}_G - x$ which implies $g(x) > g(2\widehat{\gamma}_G - x)$. Therefore $\nabla v(\widehat{\theta}_i + x) > 0$ for $x \in (2, \widehat{\gamma}_G)$. Also $\nabla v(\widehat{\theta}_i + 2\widehat{\gamma}_G - x) = -\nabla v(\widehat{\theta}_i + x)$ and therefore $\nabla v(\widehat{\theta}_i + x) < 0$ for $x \in (\widehat{\gamma}_G, 2\widehat{\gamma}_G - 2)$. $\nabla v(\widehat{\theta}_i + \widehat{\gamma}_G) = 0$ but this will be a local maxima for $\widehat{\gamma}_G > 2$ as shown in Lemma A.3. Thus there exists no local minima of $v(w)$ for $w \in (\widehat{\theta}_i + 2, \widehat{\theta}_j - 2)$

Combining both cases we see that any local minima of $v(w)$ lies in the range $\left\{ (\widehat{\theta}_i, \widehat{\theta}_i + 2] \cup [\widehat{\theta}_j - 2, \widehat{\theta}_j) \right\}$.

**Theorem A.1** *Let $w$ be a local minima of the MAXFL objective. The expected GM-Appeal using $w$ is lower bounded by $\frac{1}{16} \exp\left(-\frac{1}{\gamma^2}\right)$ where the expectation is over the randomness in the local dataset $\mathcal{B}_1, \mathcal{B}_2$.*

**Proof.**

The GM-Appeal can be written as,

$$\frac{1}{2} \left[ \mathbb{P}\left((w - \theta_i)^2 < (\widehat{\theta}_i - \theta_i)^2\right) + \mathbb{P}\left((w - \theta_j)^2 < (\widehat{\theta}_j - \theta_j)^2\right) \right] \tag{33}$$

We focus on the case where $\widehat{\theta}_2 \neq \widehat{\theta}_i$ implying $\widehat{\theta}_j > \widehat{\theta}_i$ ($\widehat{\theta}_2 = \widehat{\theta}_1$ is a zero-probability event and does not affect our proof). Let $w$ be any local minima of the MAXFL objective. From Lemma A.4 we know that $w$ will lie in the range $(\widehat{\theta}_i, \widehat{\theta}_i + 2] \cup [\widehat{\theta}_j - 2, \widehat{\theta}_j)$

**Case 1:** $w \in (\widehat{\theta}_i, \widehat{\theta}_i + 2]$

$$\mathbb{P}\left((w - \theta_i)^2 < (\widehat{\theta}_i - \theta_i)^2\right) = \mathbb{P}\left((w - \widehat{\theta}_i)^2 + 2(w - \widehat{\theta}_i)(\widehat{\theta}_i - \theta_i) < 0\right) \tag{34}$$

$$= \mathbb{P}\left((w - \widehat{\theta}_i) + 2(\widehat{\theta}_i - \theta_i) < 0\right) \tag{35}$$

$$\geq \mathbb{P}\left(2 + 2(\widehat{\theta}_i - \theta_i) < 0\right) \tag{36}$$

$$= \mathbb{P}\left((\widehat{\theta}_i - \theta_i) < -1\right) \tag{37}$$

$$\geq \mathbb{P}\left(\left\{\widehat{\theta}_1 < \widehat{\theta}_2\right\} \cap \left\{(\widehat{\theta}_1 - \theta_1) < -1\right\}\right) \tag{38}$$

$$= \mathbb{P}\left(\widehat{\theta}_1 < \widehat{\theta}_2\right) \mathbb{P}\left(\widehat{\theta}_1 - \theta_1 < -1 | \widehat{\theta}_1 < \widehat{\theta}_2\right) \tag{39}$$

$$\geq \mathbb{P}\left(\widehat{\theta}_1 < \widehat{\theta}_2\right) \mathbb{P}\left(\widehat{\theta}_1 - \theta_1 < -1\right) \tag{40}$$

$$= \mathbb{P}\left(\widehat{\theta}_1 < \widehat{\theta}_2\right) \mathbb{P}\left(Z > 1/\gamma\right) \quad \text{where } Z \sim \mathcal{N}(0, 1) \tag{41}$$

$$\geq \frac{1}{8} \exp\left(-\frac{1}{\gamma^2}\right) \tag{42}$$

(35) uses the fact that $(w - \widehat{\theta}_i) > 0$, (36) uses $(w - \widehat{\theta}_i) \leq 2$, (38) uses $\mathbb{P}(A) \geq \mathbb{P}(A \cap B)$ and definition of $i$. (40) uses the following argument. If $\theta_1 - 1 \geq \widehat{\theta}_2$ then $\mathbb{P}\left(\widehat{\theta}_1 - \theta_1 < -1 | \widehat{\theta}_1 < \widehat{\theta}_2\right) = 1$. If $\theta_1 - 1 < \widehat{\theta}_2$ then $\mathbb{P}\left(\widehat{\theta}_1 - \theta_1 < -1 | \widehat{\theta}_1 < \widehat{\theta}_2\right) = \mathbb{P}\left(\widehat{\theta}_1 - \theta_1 < -1\right) / \mathbb{P}\left(\widehat{\theta}_1 < \widehat{\theta}_2\right) \geq \mathbb{P}\left(\widehat{\theta}_1 - \theta_1 < -1\right)$. (41) uses $\widehat{\theta}_1 - \theta_1 \sim \mathcal{N}(0, \gamma^2)$, (42) uses $\mathbb{P}\left(\widehat{\theta}_1 < \widehat{\theta}_2\right) \geq \frac{1}{2}$ and $\mathbb{P}(Z \geq x) \geq \frac{2 \exp(-x^2/2)}{\sqrt{2\pi}(\sqrt{4 + x^2} + x)} \geq \frac{1}{4} \exp(-x^2)$ where $Z \sim \mathcal{N}(0, 1)$ (Komatu, 1955).

In the case where $w \in (\widehat{\theta}_j - 2, \widehat{\theta}_j]$ a similar technique can be used to lower bound $\mathbb{P}\left((w - \theta_j)^2 < (\widehat{\theta}_j - \theta_j)^2\right)$. Thus the GM-Appeal of any local minima of the MAXFL objective is lower bounded by $\frac{1}{16} \exp\left(-\frac{1}{\gamma^2}\right)$.

## C  Convergence Proof

### C.1  Preliminaries

First, we introduce the key lemmas used for the convergence analysis.

**Lemma C.1** (Bounded Dissimilarity for $\widetilde{F}(\mathbf{w})$). *With Assumption 3.1 and Assumption 3.3 we have the bounded dissimilarity with respect to $\widetilde{F}(\mathbf{w})$ as:*

$$\frac{1}{M} \sum_{i=1}^{M} \|\nabla \widetilde{F}_i(\mathbf{w})\|^2 \leq \beta'^2 \|\nabla \widetilde{F}(\mathbf{w})\|^2 + \kappa'^2 \tag{43}$$

*where $\beta'^2 = 2\beta^2$, $\kappa'^2 = 4\beta^2 L_c^2 + \kappa^2$*

*Proof.* One can easily show that

$$\frac{1}{M} \sum_{i=1}^{M} \|\nabla \widetilde{F}_i(\mathbf{w})\|^2 = \frac{1}{M} \sum_{i=1}^{M} q_i(\mathbf{w})^2 \|\nabla F_i(\mathbf{w})\|^2 \leq \frac{1}{M} \sum_{i=1}^{M} \|\nabla F_i(\mathbf{w})\|^2 \tag{44}$$

due to $q_i(\mathbf{w}) \leq 1$. Hence we have from Assumption 3.3 and Cauchy-Schwarz inequality that

$$\frac{1}{M} \sum_{i=1}^{M} \|\nabla \widetilde{F}_i(\mathbf{w})\|^2 \leq \frac{1}{M} \sum_{i=1}^{M} \|\nabla F_i(\mathbf{w})\|^2 \tag{45}$$

$$\leq \beta^2 \|\nabla F(\mathbf{w}) - \nabla \widetilde{F}(\mathbf{w}) + \nabla \widetilde{F}(\mathbf{w})\|^2 + \kappa^2 \tag{46}$$

$$\leq 2\beta^2 \|\nabla F(\mathbf{w}) - \nabla \widetilde{F}(\mathbf{w})\|^2 + 2\beta^2 \|\nabla \widetilde{F}(\mathbf{w})\|^2 + \kappa^2 \tag{47}$$

We bound the first term in (47) as

$$\|\nabla F(\mathbf{w}) - \nabla \widetilde{F}(\mathbf{w})\|^2 = \left\| \sum_{i=1}^{M} \frac{(1 - q_i(\mathbf{w}))}{M} \nabla F_i(\mathbf{w}) \right\|^2 \tag{48}$$

$$\leq \frac{1}{M} \sum_{i=1}^{M} \|(1 - q_i(\mathbf{w})) \nabla F_i(\mathbf{w})\|^2 \tag{49}$$

$$\leq \frac{2}{M} \sum_{i=1}^{M} \|\nabla F_i(\mathbf{w})\|^2 \leq 2L_c^2 \tag{50}$$

where in (50) we use $q_i(\mathbf{w}) \leq 1, \forall i \in [M]$ and Assumption 3.1. Then from (47) we have

$$\frac{1}{M} \sum_{i=1}^{M} \|\nabla \widetilde{F}_i(\mathbf{w})\|^2 \leq 2\beta^2 \|\nabla \widetilde{F}(\mathbf{w})\|^2 + \kappa^2 + 4\beta^2 L_c^2 \tag{51}$$

completing the proof. $\square$

**Lemma C.2** (Smoothness of $\widetilde{F}(\mathbf{w})$). *If Assumption 3.1 is satisfied we have that the local objectives, $\widetilde{F}_1(\mathbf{w}), \dots, \widetilde{F}_M(\mathbf{w})$, are also $\widetilde{L}_s$-smooth for any $\mathbf{w}$ where $\widetilde{L}_s = L_c^2/4 + q_i(\mathbf{w})L_s$.*

*Proof.* Recall the definitions of $\widetilde{F}(\mathbf{w})$ below:

$$\widetilde{F}(\mathbf{w}) = \frac{1}{M}\sum_{i=1}^{M}\widetilde{F}_i(\mathbf{w}), \ \ \widetilde{F}_i(\mathbf{w}) := \sigma(F_i(\mathbf{w}) - F_i(\widehat{\mathbf{w}}_i^*)) \tag{52}$$

Let $\|\ \|_{op}$ denote the spectral norm of a matrix. Accordingly, with the model parameter vector $\mathbf{w} \in \mathbb{R}^d$, we have the spectral norm of the Hessian of $\widetilde{F}_i(\mathbf{w})$, $\forall i \in [M]$ as:

$$\begin{aligned}
&\|\nabla^2 \widetilde{F}_i(\mathbf{w})\|_{op} \\
&= \|q_i(\mathbf{w})[(\nabla F_i(\mathbf{w})\nabla F_i(\mathbf{w})^T)(1 - q_i(\mathbf{w})) + \nabla^2 F_i(\mathbf{w})]\|_{op}
\end{aligned} \tag{53}$$

where $q_i(\mathbf{w}) = \text{Sigmoid}(F_i(\mathbf{w}) - F_i(\widehat{\mathbf{w}}_i^*))$ and $\nabla F_i(\mathbf{w}) \in \mathbb{R}^{d\times 1}$ is the gradient vector for the local objective $F_i(\mathbf{w})$ and $\nabla^2 F_i(\mathbf{w}) \in \mathbb{R}^{d\times d}$ is the Hessian of $F_i(\mathbf{w})$. We can bound the RHS of (53) as follows

$$\|\nabla^2 \widetilde{F}_i(\mathbf{w})\|_{op} = \|q_i(\mathbf{w})(1 - q_i(\mathbf{w}))(\nabla F_i(\mathbf{w})\nabla F_i(\mathbf{w})^T) + q_i(\mathbf{w})\nabla^2 F_i(\mathbf{w})]\|_{op} \tag{54}$$

$$\leq \|q_i(\mathbf{w})(1 - q_i(\mathbf{w}))(\nabla F_i(\mathbf{w})\nabla F_i(\mathbf{w})^T)\|_{op} + \|q_i(\mathbf{w})\nabla^2 F_i(\mathbf{w})\|_{op} \tag{55}$$

$$= q_i(\mathbf{w})(1 - q_i(\mathbf{w}))\|(\nabla F_i(\mathbf{w})\nabla F_i(\mathbf{w})^T)\|_{op} + q_i(\mathbf{w})\|\nabla^2 F_i(\mathbf{w})\|_{op} \tag{56}$$

$$= q_i(\mathbf{w})(1 - q_i(\mathbf{w}))\|\nabla F_i(\mathbf{w})\|^2 + q_i(\mathbf{w})\|\nabla^2 F_i(\mathbf{w})\|_{op} \tag{57}$$

$$\leq \frac{L_c^2}{4} + q_i(\mathbf{w})L_s \tag{58}$$

where we use triangle inequality in (55), and use $\|\mathbf{x}\mathbf{y}^T\|_{op} = \|\mathbf{x}\|\|\mathbf{y}\|$ in (57), and use $q_i(\mathbf{w}) \leq 1$ along with Assumption 3.1 in (58). Since the norm of the Hessian of $\widetilde{F}_i(\mathbf{w})$ is bounded by $\frac{L_c^2}{4} + q_i(\mathbf{w})L_s$ we complete the proof. □

## C.2 Proof of Theorem 3.1 – Full Client Participation

**Theorem C.1** (Convergence to the MAXFL Objective $\widetilde{F}(\mathbf{w})$ for Full Client Participation). *Under Assumption 3.1-3.3, suppose all $M$ clients participate for each communication round of Algorithm 1. With $\eta_l = \frac{1}{\sqrt{T}\tau}$, $\eta_g = \sqrt{\tau M}$, for total communication rounds $T$ of the MAXFL solver in Algorithm 1 we have:*

$$\begin{aligned}
\min_{t\in[T]} \mathbb{E}\left[\left\|\nabla\widetilde{F}(\mathbf{w}^{(t,0)})\right\|^2\right] &\leq \frac{(4L_s + L_c)\left(\widetilde{F}(\mathbf{w}^{(0,0)}) - \widetilde{F}_{inf}\right)}{L_s\sqrt{TM\tau}} + \frac{64L_s^2\kappa'^2(4L_s + L_c)}{L_cT} \\
&+ \frac{4L_s^2\sigma_g^2(4L_s + L_c)}{T\tau L_c} + \frac{64L_s\sigma_g^2(L_s + L_c/4)^2}{\sqrt{TM\tau}}
\end{aligned} \tag{59}$$

For ease of writing, we define the following auxiliary variables for any client $i \in [M]$:

$$\text{Weighted Stochastic Gradient: } \mathbf{h}_i^{(t,0)} := q_i(\mathbf{w}^{(t,0)})\sum_{r=0}^{\tau-1}\mathbf{g}(\mathbf{w}_i^{(t,r)}, \xi_i^{(t,r)}), \tag{60}$$

$$\text{Weighted Gradient: } \overline{\mathbf{h}}_i^{(t,0)} := q_i(\mathbf{w}^{(t,0)})\sum_{r=0}^{\tau-1}\nabla F_i(\mathbf{w}_i^{(t,r)}), \tag{61}$$

$$\text{Normalized Global Learning Rate: } \eta_g^{(t,0)} := \eta_g / \left(\sum_{i=1}^{M}q_i(\mathbf{w}^{(t,0)}) + \epsilon\right) \tag{62}$$

where $\epsilon$ is a constant added to the denominator to prevent the denominator from being 0. From Algorithm 1 with full client participation, our proposed algorithm has the following effective update rule for the global model at the server:

$$\mathbf{w}^{(t+1,0)} = \mathbf{w}^{(t,0)} - \eta_g^{(t,0)}\eta_l\sum_{k=1}^{M}\mathbf{h}_k^{(t,0)} \tag{63}$$

With the update rule in (63), defining $\widetilde{\eta}^{(t,0)} := \eta_g^{(t,0)}\eta_l\tau M$ and using Lemma C.2 we have

$$
\mathbb{E}\left[\widetilde{F}(\mathbf{w}^{(t+1,0)})\right] - \widetilde{F}(\mathbf{w}^{(t,0)}) \leq -\widetilde{\eta}^{(t,0)}\mathbb{E}\left[\left\langle \nabla\widetilde{F}(\mathbf{w}^{(t,0)}), \frac{1}{M\tau}\sum_{i=1}^{M}\mathbf{h}_i^{(t,0)} \right\rangle\right]
$$
$$
+ \frac{\widetilde{L}_s(\widetilde{\eta}^{(t,0)})^2}{2}\mathbb{E}\left[\left\|\frac{1}{M\tau}\sum_{i=1}^{M}\mathbf{h}_i^{(t,0)}\right\|^2\right] \tag{64}
$$

$$
= -\widetilde{\eta}^{(t,0)}\mathbb{E}\left[\left\langle \nabla\widetilde{F}(\mathbf{w}^{(t,0)}), \frac{1}{M\tau}\sum_{i=1}^{M}\left(\mathbf{h}_i^{(t,0)} - \overline{\mathbf{h}}_i^{(t,0)}\right) \right\rangle\right] - \widetilde{\eta}^{(t,0)}\mathbb{E}\left[\left\langle \nabla\widetilde{F}(\mathbf{w}^{(t,0)}), \frac{1}{M\tau}\sum_{i=1}^{M}\overline{\mathbf{h}}_i^{(t,0)} \right\rangle\right]
$$
$$
+ \frac{\widetilde{L}_s(\widetilde{\eta}^{(t,0)})^2}{2}\mathbb{E}\left[\left\|\frac{1}{M\tau}\sum_{i=1}^{M}\mathbf{h}_i^{(t,0)}\right\|^2\right] \tag{65}
$$

$$
= -\frac{\widetilde{\eta}^{(t,0)}}{2}\left\|\nabla\widetilde{F}(\mathbf{w}^{(t,0)})\right\|^2 - \frac{\widetilde{\eta}^{(t,0)}}{2}\mathbb{E}\left[\left\|\frac{1}{M\tau}\sum_{i=1}^{M}\overline{\mathbf{h}}_i^{(t,0)}\right\|^2\right] + \frac{\widetilde{\eta}^{(t,0)}}{2}\mathbb{E}\left[\left\|\nabla\widetilde{F}(\mathbf{w}^{(t,0)}) - \frac{1}{M\tau}\sum_{i=1}^{M}\overline{\mathbf{h}}_i^{(t,0)}\right\|^2\right]
$$
$$
+ \frac{\widetilde{L}_s(\widetilde{\eta}^{(t,0)})^2}{2M^2\tau^2}\mathbb{E}\left[\left\|\sum_{i=1}^{M}\mathbf{h}_i^{(t,0)}\right\|^2\right] \tag{66}
$$

For the last term in (66), we can bound it as

$$
\frac{\widetilde{L}_s(\widetilde{\eta}^{(t,0)})^2}{2M^2\tau^2}\mathbb{E}\left[\left\|\sum_{i=1}^{M}\mathbf{h}_i^{(t,0)}\right\|^2\right] \leq \frac{\widetilde{L}_s(\widetilde{\eta}^{(t,0)})^2}{M^2\tau^2}\sum_{i=1}^{M}\mathbb{E}\left[\left\|\mathbf{h}_i^{(t,0)} - \overline{\mathbf{h}}_i^{(t,0)}\right\|^2\right] + \frac{\widetilde{L}_s(\widetilde{\eta}^{(t,0)})^2}{M^2\tau^2}\mathbb{E}\left[\left\|\sum_{i=1}^{M}\overline{\mathbf{h}}_i^{(t,0)}\right\|^2\right] \tag{67}
$$

$$
= \frac{\widetilde{L}_s(\widetilde{\eta}^{(t,0)})^2}{M^2\tau^2}\sum_{i=1}^{M}\mathbb{E}\left[\left\|q_i(\mathbf{w}^{(t,0)})\sum_{r=0}^{\tau-1}\left(\mathbf{g}(\mathbf{w}_i^{(t,r)}, \xi_i^{(t,r)}) - \nabla F_i(\mathbf{w}_i^{(t,r)})\right)\right\|^2\right] + \frac{\widetilde{L}_s(\widetilde{\eta}^{(t,0)})^2}{M^2\tau^2}\mathbb{E}\left[\left\|\sum_{i=1}^{M}\overline{\mathbf{h}}_i^{(t,0)}\right\|^2\right] \tag{68}
$$

$$
= \frac{\widetilde{L}_s(\widetilde{\eta}^{(t,0)})^2}{M^2\tau^2}\sum_{i=1}^{M}q_i(\mathbf{w}^{(t,0)})^2\sum_{r=0}^{\tau-1}\mathbb{E}\left[\left\|\mathbf{g}(\mathbf{w}_i^{(t,r)}, \xi_i^{(t,r)}) - \nabla F_i(\mathbf{w}_i^{(t,r)})\right\|^2\right] + \frac{\widetilde{L}_s(\widetilde{\eta}^{(t,0)})^2}{M^2\tau^2}\mathbb{E}\left[\left\|\sum_{i=1}^{M}\overline{\mathbf{h}}_i^{(t,0)}\right\|^2\right] \tag{69}
$$

$$
= \frac{\widetilde{L}_s(\widetilde{\eta}^{(t,0)})^2}{M^2\tau^2}\sum_{i=1}^{M}q_i(\mathbf{w}^{(t,0)})^2\tau\sigma_g^2 + \frac{\widetilde{L}_s(\widetilde{\eta}^{(t,0)})^2}{M^2\tau^2}\mathbb{E}\left[\left\|\sum_{i=1}^{M}\overline{\mathbf{h}}_i^{(t,0)}\right\|^2\right] \tag{70}
$$

$$
\leq \frac{\widetilde{L}_s(\widetilde{\eta}^{(t,0)})^2\sigma_g^2}{M\tau} + \widetilde{L}_s(\widetilde{\eta}^{(t,0)})^2\mathbb{E}\left[\left\|\frac{1}{M\tau}\sum_{i=1}^{M}\overline{\mathbf{h}}_i^{(t,0)}\right\|^2\right] \tag{71}
$$

where (67) is due to the Cauchy-Schwartz inequality and (70) is due to Assumption 3.2 and (71) is due to $q_i(\mathbf{w}) \leq 1, \forall i \in [M]$. Merging (71) into (66) we have

$$
\mathbb{E}\left[\widetilde{F}(\mathbf{w}^{(t+1,0)})\right] - \widetilde{F}(\mathbf{w}^{(t,0)}) \leq -\frac{\widetilde{\eta}^{(t,0)}}{2}\left\|\nabla\widetilde{F}(\mathbf{w}^{(t,0)})\right\|^2 + \frac{\widetilde{\eta}^{(t,0)}}{2}\mathbb{E}\left[\left\|\nabla\widetilde{F}(\mathbf{w}^{(t,0)}) - \frac{1}{M\tau}\sum_{i=1}^{M}\overline{\mathbf{h}}_i^{(t,0)}\right\|^2\right]
$$
$$
+ \frac{\widetilde{L}_s(\widetilde{\eta}^{(t,0)})^2\sigma_g^2}{M\tau} + \left((\widetilde{\eta}^{(t,0)})^2\widetilde{L}_s - \frac{\widetilde{\eta}^{(t,0)}}{2}\right)\mathbb{E}\left[\left\|\frac{1}{M\tau}\sum_{i=1}^{M}\overline{\mathbf{h}}_i^{(t,0)}\right\|^2\right] \tag{72}
$$

Now we aim at bounding the second term in the RHS of (72) as follows:

$$\frac{\widetilde{\eta}^{(t,0)}}{2}\mathbb{E}\left[\left\|\nabla\widetilde{F}(\mathbf{w}^{(t,0)}) - \frac{1}{M\tau}\sum_{i=1}^{M}\overline{\mathbf{h}}_i^{(t,0)}\right\|^2\right] \tag{73}$$

$$= \frac{\widetilde{\eta}^{(t,0)}}{2}\mathbb{E}\left[\left\|\frac{1}{M}\sum_{i=1}^{M}q_i(\mathbf{w}^{(t,0)})\nabla F_i(\mathbf{w}^{(t,0)}) - \frac{1}{M\tau}\sum_{i=1}^{M}q_i(\mathbf{w}^{(t,0)})\sum_{r=0}^{\tau-1}\nabla F_i(\mathbf{w}_i^{(t,r)})\right\|^2\right] \tag{74}$$

$$= \frac{\widetilde{\eta}^{(t,0)}}{2}\mathbb{E}\left[\left\|\frac{1}{M\tau}\sum_{i=1}^{M}q_i(\mathbf{w}^{(t,0)})\sum_{r=0}^{\tau-1}\left(\nabla F_i(\mathbf{w}^{(t,0)}) - \nabla F_i(\mathbf{w}_i^{(t,r)})\right)\right\|^2\right] \tag{75}$$

$$\leq \frac{\widetilde{\eta}^{(t,0)}}{2M\tau}\sum_{i=1}^{M}q_i(\mathbf{w}^{(t,0)})^2\sum_{r=0}^{\tau-1}\mathbb{E}\left[\left\|\nabla F_i(\mathbf{w}^{(t,0)}) - \nabla F_i(\mathbf{w}_i^{(t,r)})\right\|^2\right] \tag{76}$$

$$= \frac{L_s^2\widetilde{\eta}^{(t,0)}}{2M\tau}\sum_{i=1}^{M}q_i(\mathbf{w}^{(t,0)})^2\sum_{r=0}^{\tau-1}\mathbb{E}\left[\left\|\mathbf{w}^{(t,0)} - \mathbf{w}_i^{(t,r)}\right\|^2\right] \tag{77}$$

where (76) is due to Jensen's inequality and (77) is due to Lemma C.2. We can bound the difference of the global model and local model for any client $i \in [M]$ as follows:

$$\mathbb{E}\left[\left\|\mathbf{w}^{(t,0)} - \mathbf{w}_i^{(t,r)}\right\|^2\right] = \eta_l^2\mathbb{E}\left[\left\|\sum_{l=0}^{r-1}\mathbf{g}(\mathbf{w}_i^{(t,l)},\xi_i^{(t,l)})\right\|^2\right] \tag{78}$$

$$\leq 2\eta_l^2\mathbb{E}\left[\left\|\sum_{l=0}^{r-1}\mathbf{g}(\mathbf{w}_i^{(t,l)},\xi_i^{(t,l)}) - \nabla F_i(\mathbf{w}_i^{(t,l)})\right\|^2\right] + 2\eta_l^2\mathbb{E}\left[\left\|\sum_{l=0}^{r-1}\nabla F_i(\mathbf{w}_i^{(t,l)})\right\|^2\right] \tag{79}$$

$$\leq 2\eta_l^2\sigma_g^2 r + 2\eta_l^2\mathbb{E}\left[\left\|\sum_{l=0}^{r-1}\nabla F_i(\mathbf{w}_i^{(t,l)})\right\|^2\right] \tag{80}$$

where (79) is due to Cauchy-Schwarz inequality and (80) is due to Assumption 3.2. We bound the last term in (80) as follows:

$$\mathbb{E}\left[\left\|\sum_{l=0}^{r-1}\nabla F_i(\mathbf{w}_i^{(t,l)})\right\|^2\right] \leq r\sum_{l=0}^{r-1}\mathbb{E}\left[\left\|\nabla F_i(\mathbf{w}_i^{(t,l)})\right\|^2\right] \leq \tau\sum_{l=0}^{\tau-1}\mathbb{E}\left[\left\|\nabla F_i(\mathbf{w}_i^{(t,l)})\right\|^2\right] \tag{81}$$

$$\leq 2\tau\sum_{l=0}^{\tau-1}\mathbb{E}\left[\left\|\nabla F_i(\mathbf{w}_i^{(t,l)}) - \nabla F_i(\mathbf{w}^{(t,0)})\right\|^2\right] + 2\tau^2\mathbb{E}\left[\left\|\nabla F_i(\mathbf{w}^{(t,0)})\right\|^2\right] \tag{82}$$

$$\leq 2L_s^2\tau\sum_{l=0}^{\tau-1}\mathbb{E}\left[\left\|\mathbf{w}_i^{(t,l)} - \mathbf{w}^{(t,0)}\right\|^2\right] + 2\tau^2\mathbb{E}\left[\left\|\nabla F_i(\mathbf{w}^{(t,0)})\right\|^2\right] \tag{83}$$

where (81) is due to Jensen's inequality, and (82) is due to Cauchy-Schwarz inequality, and (83) is due to Lemma C.2. Combining (83) with (80) we have that

$$\mathbb{E}\left[\left\|\mathbf{w}^{(t,0)} - \mathbf{w}_i^{(t,r)}\right\|^2\right] \leq 2\eta_l^2\sigma_g^2 r + 4L_s^2\eta_l^2\tau\sum_{l=0}^{\tau-1}\mathbb{E}\left[\left\|\mathbf{w}^{(t,0)} - \mathbf{w}_i^{(t,l)}\right\|^2\right] + 4\eta_l^2\tau^2\mathbb{E}\left[\left\|\nabla F_i(\mathbf{w}^{(t,0)})\right\|^2\right] \tag{84}$$

Reorganizing (84) and taking the summation $r \in [\tau]$ on both sides we have,

$$(1 - 4L_s^2\eta_l^2\tau^2)\sum_{r=0}^{\tau-1}\mathbb{E}\left[\left\|\mathbf{w}^{(t,0)} - \mathbf{w}_i^{(t,r)}\right\|^2\right] \leq 2\eta_l^2\sigma_g^2\sum_{r=0}^{\tau-1}r + 4\eta_l^2\tau^3\mathbb{E}\left[\left\|\nabla F_i(\mathbf{w}^{(t,0)})\right\|^2\right] \tag{85}$$

$$\leq \eta_l^2\sigma_g^2\tau^2 + 4\eta_l^2\tau^3\mathbb{E}\left[\left\|\nabla F_i(\mathbf{w}^{(t,0)})\right\|^2\right] \tag{86}$$

With $\eta_l \le 1/(2\sqrt{2}\tau L_s)$, we have that $1/(1 - 4L_s^2\eta_l^2\tau^2) \le 2$ and hence can further bound (86) as

$$\sum_{r=0}^{\tau-1} \mathbb{E}\left[\left\|\mathbf{w}^{(t,0)} - \mathbf{w}_i^{(t,r)}\right\|^2\right] \le 2\eta_l^2\sigma_g^2\tau^2 + 8\eta_l^2\tau^3\mathbb{E}\left[\left\|\nabla F_i(\mathbf{w}^{(t,0)})\right\|^2\right] \tag{87}$$

Finally, plugging in (87) to (77) we have

$$\frac{\widetilde{\eta}^{(t,0)}}{2}\mathbb{E}\left[\left\|\nabla\widetilde{F}(\mathbf{w}^{(t,0)}) - \frac{1}{M\tau}\sum_{i=1}^{M}\overline{\mathbf{h}}_i^{(t,0)}\right\|^2\right] \tag{88}$$

$$\le \frac{L_s^2\widetilde{\eta}^{(t,0)}}{2M\tau}\sum_{i=1}^{M} q_i(\mathbf{w}^{(t,0)})^2\left(2\eta_l^2\sigma_g^2\tau^2 + 8\eta_l^2\tau^3\mathbb{E}\left[\left\|\nabla F_i(\mathbf{w}^{(t,0)})\right\|^2\right]\right)$$

$$\le L_s^2\widetilde{\eta}^{(t,0)}\eta_l^2\sigma_g^2\tau + 4\eta_l^2\tau^2 L_s^2\widetilde{\eta}^{(t,0)}\frac{1}{M}\sum_{i=1}^{M}\mathbb{E}\left[\left\|\nabla F_i(\mathbf{w}^{(t,0)})\right\|^2\right] \tag{89}$$

$$\le L_s^2\widetilde{\eta}^{(t,0)}\eta_l^2\sigma_g^2\tau + 4\eta_l^2\tau^2 L_s^2\widetilde{\eta}^{(t,0)}(\beta'^2\left\|\nabla\widetilde{F}(\mathbf{w}^{(t,0)})\right\|^2 + \kappa'^2) \tag{90}$$

where (89) uses $q_i(\mathbf{w}) \le 1, \forall i \in [M]$ and (90) uses Lemma C.1. Merging (90) to (72) we have

$$\mathbb{E}\left[\widetilde{F}(\mathbf{w}^{(t+1,0)})\right] - \widetilde{F}(\mathbf{w}^{(t,0)})$$

$$\le -\frac{\widetilde{\eta}^{(t,0)}}{2}\left\|\nabla\widetilde{F}(\mathbf{w}^{(t,0)})\right\|^2 + \widetilde{\eta}^{(t,0)}\left(\widetilde{\eta}^{(t,0)}\widetilde{L}_s - \frac{1}{2}\right)\mathbb{E}\left[\left\|\frac{1}{M\tau}\sum_{i=1}^{M}\overline{\mathbf{h}}_i^{(t,0)}\right\|^2\right] \tag{91}$$

$$+ \frac{\widetilde{L}_s(\widetilde{\eta}^{(t,0)})^2\sigma_g^2}{M\tau} + \widetilde{\eta}^{(t,0)}L_s^2\eta_l^2\sigma_g^2\tau + 4\widetilde{\eta}^{(t,0)}\eta_l^2\tau^2 L_s^2\beta'^2\left\|\nabla\widetilde{F}(\mathbf{w}^{(t,0)})\right\|^2 + 4\widetilde{\eta}^{(t,0)}\eta_l^2\tau^2 L_s^2\kappa'^2$$

With $\eta_l\eta_g \le 1/(4\tau L_s)$ we have that $\widetilde{\eta}^{(t,0)}\widetilde{L}_s - \frac{1}{2} \le -1/4$ and thus can further simplify (91) to

$$\mathbb{E}\left[\widetilde{F}(\mathbf{w}^{(t+1,0)})\right] - \widetilde{F}(\mathbf{w}^{(t,0)}) \le -\frac{\widetilde{\eta}^{(t,0)}}{2}\left\|\nabla\widetilde{F}(\mathbf{w}^{(t,0)})\right\|^2 + 4\widetilde{\eta}^{(t,0)}\eta_l^2\tau^2 L_s^2\beta'^2\left\|\nabla\widetilde{F}(\mathbf{w}^{(t,0)})\right\|^2$$

$$+ \frac{\widetilde{L}_s(\widetilde{\eta}^{(t,0)})^2\sigma_g^2}{M\tau} + \widetilde{\eta}^{(t,0)}L_s^2\eta_l^2\sigma_g^2\tau + 4\widetilde{\eta}^{(t,0)}\eta_l^2\tau^2 L_s^2\kappa'^2 \tag{92}$$

$$= \widetilde{\eta}^{(t,0)}\left(4\eta_l^2\tau^2 L_s^2\beta' - \frac{1}{2}\right)\left\|\nabla\widetilde{F}(\mathbf{w}^{(t,0)})\right\|^2 + \frac{\widetilde{L}_s(\widetilde{\eta}^{(t,0)})^2\sigma_g^2}{M\tau} + \widetilde{\eta}^{(t,0)}L_s^2\eta_l^2\sigma_g^2\tau + 4\widetilde{\eta}^{(t,0)}\eta_l^2\tau^2 L_s^2\kappa'^2 \tag{93}$$

With local learning rate $\eta_l \le \min\{1/(4\tau L_s), 1/(4\beta'\tau L_s)\}$ we have that

$$\mathbb{E}\left[\widetilde{F}(\mathbf{w}^{(t+1,0)})\right] - \widetilde{F}(\mathbf{w}^{(t,0)}) \le -\frac{\widetilde{\eta}^{(t,0)}}{4}\left\|\nabla\widetilde{F}(\mathbf{w}^{(t,0)})\right\|^2 + \frac{\widetilde{L}_s(\widetilde{\eta}^{(t,0)})^2\sigma_g^2}{M\tau} + \widetilde{\eta}^{(t,0)}L_s^2\eta_l^2\sigma_g^2\tau$$

$$+ 4\widetilde{\eta}^{(t,0)}\eta_l^2\tau^2 L_s^2\kappa'^2 \tag{94}$$

and we use the property of $\widetilde{\eta}^{(t,0)}$ that $\frac{M\tau\eta_l\eta_g}{M+\epsilon} \le \widetilde{\eta}^{(t,0)} \le \frac{M\tau\eta_l\eta_g}{\epsilon}$ to get

$$\mathbb{E}\left[\widetilde{F}(\mathbf{w}^{(t+1,0)})\right] - \widetilde{F}(\mathbf{w}^{(t,0)}) \le -\frac{M\tau\eta_l\eta_g}{4(M+\epsilon)}\left\|\nabla\widetilde{F}(\mathbf{w}^{(t,0)})\right\|^2 + \frac{\widetilde{L}_s M\tau\eta_l^2\eta_g^2\sigma_g^2}{\epsilon^2}$$

$$+ \frac{M\tau^2 L_s^2\eta_l^3\eta_g\sigma_g^2}{\epsilon} + \frac{4M\eta_l^3\eta_g\tau^3 L_s^2\kappa'^2}{\epsilon} \tag{95}$$

Taking the average across all rounds on both sides of (95) we get

$$\frac{1}{T}\sum_{t=0}^{T-1}\mathbb{E}\left[\|\nabla\widetilde{F}(\mathbf{w}^{(t,0)})\|^2\right] \le \frac{4(M+\epsilon)\left(\widetilde{F}(\mathbf{w}^{(0,0)}) - \widetilde{F}_{\inf}\right)}{M\tau\eta_l\eta_g T} + \frac{16\eta_l^2\tau^2 L_s^2\kappa'^2(M+\epsilon)}{\epsilon}$$

$$+ \frac{4L_s^2\eta_l^2\tau\sigma_g^2(M+\epsilon)}{\epsilon} + \frac{4\eta_g\eta_l\widetilde{L}_s\sigma_g^2(M+\epsilon)}{\epsilon^2} \tag{96}$$

and prove

$$
\min_{t \in [T]} \mathbb{E}\left[\left\|\nabla \widetilde{F}(\mathbf{w}^{(t,0)})\right\|^2\right] \leq \frac{1}{T} \sum_{t=0}^{T-1} \mathbb{E}\left[\|\nabla \widetilde{F}(\mathbf{w}^{(t,0)})\|^2\right] \leq \frac{4(M+\epsilon)\left(\widetilde{F}(\mathbf{w}^{(0,0)}) - \widetilde{F}_{\inf}\right)}{M\tau\eta_l\eta_g T}
$$
$$
+ \frac{16\eta_l^2\tau^2 L_s^2\kappa'^2(M+\epsilon)}{\epsilon} + \frac{4L_s^2\eta_l^2\tau\sigma_g^2(M+\epsilon)}{\epsilon} + \frac{4\eta_g\eta_l\widetilde{L}_s\sigma_g^2(M+\epsilon)}{\epsilon^2}
\tag{97}
$$

Further, using $\tilde{L}_s = \frac{L_s}{M}\sum_{k=1}^M q_k(\mathbf{w}) + \frac{L_c}{4}$ and $\epsilon = \frac{ML_c}{4L_s} > 0$ from the optimal learning rate we have the bound in (97) to be

$$
\min_{t \in [T]} \mathbb{E}\left[\left\|\nabla \widetilde{F}(\mathbf{w}^{(t,0)})\right\|^2\right] \leq \frac{(4L_s + L_c)\left(\widetilde{F}(\mathbf{w}^{(0,0)}) - \widetilde{F}_{\inf}\right)}{L_s\tau\eta_l\eta_g T} + \frac{64\eta_l^2\tau^2 L_s^2\kappa'^2(4L_s + L_c)}{L_c}
$$
$$
+ \frac{4L_s^2\eta_l^2\tau\sigma_g^2(4L_s + L_c)}{L_c} + \frac{64L_s\eta_g\eta_l\sigma_g^2(L_s + L_c/4)^2}{ML_c^2}
\tag{98}
$$

By setting the global and local learning rate as $\eta_g = \sqrt{\tau M}$ and $\eta_l = \frac{1}{\sqrt{T}\tau}$ we can further optimize the bound as

$$
\min_{t \in [T]} \mathbb{E}\left[\left\|\nabla \widetilde{F}(\mathbf{w}^{(t,0)})\right\|^2\right] \leq \frac{(4L_s + L_c)\left(\widetilde{F}(\mathbf{w}^{(0,0)}) - \widetilde{F}_{\inf}\right)}{L_s\sqrt{TM\tau}} + \frac{64L_s^2\kappa'^2(4L_s + L_c)}{L_c T}
$$
$$
+ \frac{4L_s^2\sigma_g^2(4L_s + L_c)}{T\tau L_c} + \frac{64L_s\sigma_g^2(L_s + L_c/4)^2}{\sqrt{TM\tau}}
\tag{99}
$$

completing the full client participation proof of Theorem 3.1.

## C.3 Proof of Theorem 3.1 – Partial Client Participation

**Theorem C.2** (Convergence to the MAXFL Objective $\widetilde{F}(\mathbf{w})$ for Partial Client Participation). *Under Assumption 3.1-3.3, suppose $m$ clients are selected by the server uniformly at random without replacement out of $M$ total clients for participation for each communication round of Algorithm 1. With $\eta_l = \frac{1}{\sqrt{T}\tau}$, $\eta_g = \sqrt{\tau m}$, for total communication rounds $T$ of the MAXFL solver in Algorithm 1 we have:*

$$
\min_{t \in [T]} \mathbb{E}\left[\left\|\nabla \widetilde{F}(\mathbf{w}^{(t,0)})\right\|^2\right] \leq \frac{4\left(\widetilde{F}(\mathbf{w}^{(0,0)}) - \widetilde{F}_{inf}\right) + 4\sigma_g^2\nu}{\sqrt{T\tau m}} + \frac{4\sigma_g^2 L_s^2}{\sqrt{T}} + \frac{8\sigma_g^2 L_s^2}{3\tau T}
$$
$$
+ \frac{80 L_s^2\kappa'^2}{T} + \frac{48\nu(M-m)L_c^2\sqrt{\tau}}{\sqrt{Tm}}
\tag{100}
$$

We present the convergence guarantees of MAXFL for partical client participation in this section. With partical client participation, we have the update rule in (63) changed to

$$
\mathbf{w}^{(t+1,0)} = \mathbf{w}^{(t,0)} - \eta_g^{(t,0)}\eta_l \sum_{k \in \mathcal{S}^{(t,0)}} \mathbf{h}_k^{(t,0)}
\tag{101}
$$

where the $m$ clients are sampled uniformly at random without replacement for $\mathcal{S}^{(t,0)}$ at each communication round $t$ by the server and $\eta_g^{(t,0)} = m\eta_g/(\sum_{k \in \mathcal{S}^{(t,0)}} q_k(\mathbf{w}^{(t,0)}) + \epsilon)$ for positive constant $\epsilon$. Then with the update rule in (101) and Lemma C.2, defining $\widetilde{\eta}^{(t,0)} = \eta_g^{(t,0)}\eta_l\tau m$ we have

$$
\mathbb{E}\left[\widetilde{F}(\mathbf{w}^{(t+1,0)})\right] - \widetilde{F}(\mathbf{w}^{(t,0)}) \leq \mathbb{E}\left[-\widetilde{\eta}^{(t,0)}\left\langle \nabla \widetilde{F}(\mathbf{w}^{(t,0)}), \frac{1}{m\tau}\sum_{i \in \mathcal{S}^{(t,0)}} \mathbf{h}_i^{(t,0)}\right\rangle\right]
$$
$$
+ \mathbb{E}\left[\frac{\widetilde{L}_s(\widetilde{\eta}^{(t,0)})^2}{2}\left\|\frac{1}{m\tau}\sum_{i \in \mathcal{S}^{(t,0)}} \mathbf{h}_i^{(t,0)}\right\|^2\right]
\tag{102}
$$

For the first term in the RHS of (102) we have that due to the uniform sampling of clients (see Lemma 4 in Jhunjhunwala et al. (2022)), it becomes analogous to the derivation for full client participation. Hence, with the property of $\frac{m\tau\eta_l\eta_g}{m+\epsilon} \leq \widetilde{\eta}^{(t,0)} \leq \frac{m\tau\eta_l\eta_g}{\epsilon}$ and using the previous bounds in (90), we result in the final bound for the first term in the RHS of (102) as below:

$$\mathbb{E}\left[-\widetilde{\eta}^{(t,0)}\left\langle \nabla\widetilde{F}(\mathbf{w}^{(t,0)}), \frac{1}{m\tau}\sum_{i\in\mathcal{S}^{(t,0)}}\mathbf{h}_i^{(t,0)}\right\rangle\right] \leq \left(-\frac{m\tau\eta_l\eta_g}{m+\epsilon} + \frac{4\eta_l^3\tau^3 L_s^2\beta'^2\eta_g m}{\epsilon}\right)\left\|\nabla\widetilde{F}(\mathbf{w}^{(t,0)})\right\|^2 \tag{103}$$
$$+\frac{4L_s^2\tau^3\eta_l^3 m\eta_g\kappa'^2}{\epsilon} + \frac{L_s^2\tau^2\eta_l^2 m\eta_g\sigma_g^2}{\epsilon}$$

For the second term in the RHS of (102), with $C = \widetilde{L}_s(m\tau\eta_l\eta_g/\epsilon)^2$ we have the following:

$$\mathbb{E}\left[\frac{\widetilde{L}_s(\widetilde{\eta}^{(t,0)})^2}{2}\left\|\frac{1}{m\tau}\sum_{i\in\mathcal{S}^{(t,0)}}\mathbf{h}_i^{(t,0)}\right\|^2\right] \leq C\mathbb{E}\left[\left\|\frac{1}{m\tau}\sum_{i\in\mathcal{S}^{(t,0)}}(\mathbf{h}_i^{(t,0)} - \overline{\mathbf{h}}_i^{(t,0)})\right\|^2\right]$$
$$+C\mathbb{E}\left[\left\|\frac{1}{m\tau}\sum_{i\in\mathcal{S}^{(t,0)}}\overline{\mathbf{h}}_i^{(t,0)}\right\|^2\right] \tag{104}$$

$$= \frac{C}{m^2\tau^2}\mathbb{E}\left[\sum_{i\in\mathcal{S}^{(t,0)}}\left\|\mathbf{h}_i^{(t,0)} - \overline{\mathbf{h}}_i^{(t,0)}\right\|^2\right] + C\mathbb{E}\left[\left\|\frac{1}{m\tau}\sum_{i\in\mathcal{S}^{(t,0)}}\overline{\mathbf{h}}_i^{(t,0)}\right\|^2\right] \tag{105}$$

$$= \frac{C}{mM\tau^2}\sum_{i=1}^{M}\mathbb{E}\left[\left\|\mathbf{h}_i^{(t,0)} - \overline{\mathbf{h}}_i^{(t,0)}\right\|^2\right] + C\mathbb{E}\left[\left\|\frac{1}{m\tau}\sum_{i\in\mathcal{S}^{(t,0)}}\overline{\mathbf{h}}_i^{(t,0)}\right\|^2\right] \tag{106}$$

$$\leq \frac{C\sigma_g^2}{m\tau} + C\mathbb{E}\left[\left\|\frac{1}{m\tau}\sum_{i\in\mathcal{S}^{(t,0)}}\overline{\mathbf{h}}_i^{(t,0)}\right\|^2\right] \tag{107}$$

where (106) follows due to, again, the uniform sampling of clients and the rest follows identical steps for full client participation in the derivation for (67). Note that

$$C = \left(\frac{L_s}{M}\sum_{k=1}^{M}q_k(\mathbf{w}) + \frac{L_c}{4}\right)(m\tau\eta_l\eta_g/\epsilon)^2 \leq (L_s + \frac{L_c}{4})(m\tau\eta_l\eta_g/\epsilon)^2 \tag{108}$$

For the second term in (107) we have that

$$\mathbb{E}\left[\left\|\frac{1}{m\tau}\sum_{i\in\mathcal{S}^{(t,0)}}\overline{\mathbf{h}}_i^{(t,0)}\right\|^2\right] = \mathbb{E}\left[\left\|\frac{1}{m\tau}\sum_{i\in\mathcal{S}^{(t,0)}}\left(\overline{\mathbf{h}}_i^{(t,0)} - \nabla\widetilde{F}_i(\mathbf{w}^{(t,0)}) + \nabla\widetilde{F}_i(\mathbf{w}^{(t,0)})\right)\right.\right.$$
$$\left.\left. -\frac{1}{\tau}\nabla\widetilde{F}(\mathbf{w}^{(t,0)}) + \frac{1}{\tau}\nabla\widetilde{F}(\mathbf{w}^{(t,0)})\right\|^2\right] \tag{109}$$

$$\leq \underbrace{3\mathbb{E}\left[\left\|\frac{1}{m\tau}\sum_{i\in\mathcal{S}^{(t,0)}}\left(\overline{\mathbf{h}}_i^{(t,0)} - \nabla\widetilde{F}_i(\mathbf{w}^{(t,0)})\right)\right\|^2\right]}_{A_1} + \underbrace{\frac{3}{\tau^2}\mathbb{E}\left[\left\|\frac{1}{m}\sum_{i\in\mathcal{S}^{(t,0)}}\nabla\widetilde{F}_i(\mathbf{w}^{(t,0)}) - \nabla\widetilde{F}(\mathbf{w}^{(t,0)})\right\|^2\right]}_{A_2}$$
$$+3\mathbb{E}\left[\left\|\frac{1}{\tau}\nabla\widetilde{F}(\mathbf{w}^{(t,0)})\right\|^2\right] \tag{110}$$

First we bound $A_1$ in (110) as follows:

$$3\mathbb{E}\left[\left\|\frac{1}{m\tau}\sum_{i\in\mathcal{S}^{(t,0)}}\left(\overline{\mathbf{h}}_i^{(t,0)}-\nabla\widetilde{F}_i(\mathbf{w}^{(t,0)})\right)\right\|^2\right]=3\mathbb{E}\left[\left\|\frac{1}{m\tau}\sum_{i\in\mathcal{S}^{(t,0)}}q_i(\mathbf{w}^{(t,0)})\sum_{r=0}^{\tau-1}\left(\nabla F_i(\mathbf{w}_i^{(t,r)})-\nabla F_i(\mathbf{w}^{(t,0)})\right)\right\|^2\right]$$

$$\leq\frac{3}{m\tau}\mathbb{E}\left[\sum_{i\in\mathcal{S}^{(t,0)}}\sum_{r=0}^{\tau-1}\left\|\nabla F_i(\mathbf{w}_i^{(t,r)})-\nabla F_i(\mathbf{w}^{(t,0)})\right\|^2\right]=\frac{3}{M\tau}\sum_{i=1}^{M}\sum_{r=0}^{\tau-1}\mathbb{E}\left[\left\|\nabla F_i(\mathbf{w}_i^{(t,r)})-\nabla F_i(\mathbf{w}^{(t,0)})\right\|^2\right]$$

$$\tag{111}$$

$$\leq\frac{3L_s^2}{M\tau}\sum_{i=1}^{M}\sum_{r=0}^{\tau-1}\mathbb{E}\left[\left\|\mathbf{w}^{(t,0)}-\mathbf{w}_i^{(t,r)}\right\|^2\right]\tag{112}$$

where (111) is due to Jensen's inequality, $q_i(\mathbf{w})\leq 1$, and uniform sampling of clients, and (112) is due to Assumption 3.1. Using (72) we have already derived, bound (112) further to:

$$3\mathbb{E}\left[\left\|\frac{1}{m\tau}\sum_{i\in\mathcal{S}^{(t,0)}}\left(\overline{\mathbf{h}}_i^{(t,0)}-\nabla\widetilde{F}_i(\mathbf{w}^{(t,0)})\right)\right\|^2\right]\leq 6L_s^2\eta_l^2\sigma_g^2\tau+\frac{24L_s^2\eta_l^2\tau^2}{M}\sum_{i=1}^{M}\mathbb{E}\left[\left\|\nabla F_i(\mathbf{w}^{(t,0)})\right\|^2\right]\tag{113}$$

$$\leq 6L_s^2\eta_l^2\sigma_g^2\tau+24L_s^2\eta_l^2\tau^2(\beta'^2\|\nabla\widetilde{F}(\mathbf{w}^{(t,0)})\|^2+\kappa'^2)\tag{114}$$

where (114) is due to Lemma C.1.

Next we bound $A_2$ as follows:

$$\frac{3}{\tau^2}\mathbb{E}\left[\left\|\frac{1}{m}\sum_{i\in\mathcal{S}^{(t,0)}}\nabla\widetilde{F}_i(\mathbf{w}^{(t,0)})-\nabla\widetilde{F}(\mathbf{w}^{(t,0)})\right\|^2\right]\tag{115}$$

$$=\frac{3(M-m)}{\tau^2 mM(M-1)}\sum_{i=1}^{M}\mathbb{E}\left[\left\|\nabla\widetilde{F}_i(\mathbf{w}^{(t,0)})-\nabla\widetilde{F}(\mathbf{w}^{(t,0)})\right\|^2\right]$$

$$=\frac{3(M-m)}{\tau^2 mM(M-1)}\sum_{i=1}^{M}\left\|\nabla q_i(\mathbf{w}^{(t,0)})F_i(\mathbf{w}^{(t,0)})-\frac{1}{M}\sum_{i=1}^{M}q_i(\mathbf{w}^{(t,0)})\nabla F_i(\mathbf{w}^{(t,0)})\right\|^2\tag{116}$$

$$\leq\frac{6(M-m)}{\tau^2 mM(M-1)}\sum_{i=1}^{M}\left(\left\|\nabla F_i(\mathbf{w}^{(t,0)})\right\|^2+\left\|\frac{1}{M}\sum_{i=1}^{M}q_i(\mathbf{w}^{(t,0)})\nabla F_i(\mathbf{w}^{(t,0)})\right\|^2\right)\tag{117}$$

$$\leq\frac{12(M-m)L_c^2}{\tau^2 m(M-1)}\tag{118}$$

where (115) is due to the variance under uniform sampling without replacement (see Lemma 4 in Jhunjhunwala et al. (2022)) and (117) is due to the Cauchy-Schwarz inequality and (118) is due to Assumption 3.1.

Mering the bounds for $A_1$ and $A_2$ to (110) we have that

$$\mathbb{E}\left[\left\|\frac{1}{m\tau}\sum_{i\in\mathcal{S}^{(t,0)}}\overline{\mathbf{h}}_i^{(t,0)}\right\|^2\right]\leq 6L_s^2\eta_l^2\sigma_g^2\tau+24L_s^2\eta_l^2\tau^2\beta'^2\|\nabla\widetilde{F}(\mathbf{w}^{(t,0)})\|^2$$

$$+24L_s^2\eta_l^2\tau^2\kappa'^2+\frac{12(M-m)L_c^2}{\tau^2 m(M-1)}+3\mathbb{E}\left[\left\|\frac{1}{\tau}\nabla\widetilde{F}(\mathbf{w}^{(t,0)})\right\|^2\right]\tag{119}$$

$$=\left(24L_s^2\eta_l^2\tau^2\beta'^2+\frac{3}{\tau^2}\right)\|\nabla\widetilde{F}(\mathbf{w}^{(t,0)})\|^2+6L_s^2\eta_l^2\tau(\sigma_g^2+4\tau\kappa'^2)+\frac{12(M-m)L_c^2}{\tau^2 m(M-1)}\tag{120}$$

Then we can plug in (120) back to (107) and plugging in (103) to (102), we can derive the bound in (102) as

$$
\mathbb{E}\left[\widetilde{F}(\mathbf{w}^{(t+1,0)})\right] - \widetilde{F}(\mathbf{w}^{(t,0)})
$$

$$
\leq \left(-\frac{m\tau\eta_l\eta_g}{m+\epsilon} + \frac{4\eta_l^3\eta_g\tau^3 L_s^2\beta'^2 m}{\epsilon} + \nu\left(\tau\eta_l\eta_g\right)^2\left(24L_s^2\eta_l^2\tau^2\beta'^2 + 3\right)\right)\left\|\nabla\widetilde{F}(\mathbf{w}^{(t,0)})\right\|^2 + (\tau\eta_l\eta_g)^2\nu\frac{\sigma_g^2}{m\tau} \quad (121)
$$

$$
+ (\tau\eta_l\eta_g)^2\nu\left(6L_s^2\eta_l^2\tau(\sigma_g^2 + 4\tau\kappa'^2) + \frac{12(M-m)L_c^2}{\tau^2 m(M-1)}\right) + \frac{4L_s^2\tau^3\eta_l^3 m\eta_g\kappa'^2}{\epsilon} + \frac{L_s^2\tau^2\eta_l^2 m\eta_g\sigma_g^2}{\epsilon}
$$

where $\nu = L_s + L_c/4$. With $\eta_l \leq 1/4\beta'\tau L_s$, $\epsilon = m$, and $\eta_g\eta_l \leq \frac{1}{9\tau\nu}$, we can further bound above as

$$
\mathbb{E}\left[\widetilde{F}(\mathbf{w}^{(t+1,0)})\right] - \widetilde{F}(\mathbf{w}^{(t,0)}) \leq -\frac{\eta_l\eta_g\tau}{4}\left\|\nabla\widetilde{F}(\mathbf{w}^{(t,0)})\right\|^2 + (\tau\eta_l\eta_g)^2\nu\frac{\sigma_g^2}{m\tau}
$$

$$
+ (\tau\eta_l\eta_g)^2\nu\left(6L_s^2\eta_l^2\tau(\sigma_g^2 + 4\tau\kappa'^2) + \frac{12(M-m)L_c^2}{\tau^2 m(M-1)}\right) + 4L_s^2\tau^3\eta_l^3\eta_g\kappa'^2 + L_s^2\tau^2\eta_l^2\eta_g\sigma_g^2 \quad (122)
$$

Taking the average across all rounds on both sides of (122) and rearranging the terms we get

$$
\frac{1}{T}\sum_{t=0}^{T-1}\mathbb{E}\left[\|\nabla\widetilde{F}(\mathbf{w}^{(t,0)})\|^2\right] \leq \frac{4\left(\widetilde{F}(\mathbf{w}^{(0,0)}) - \widetilde{F}_{\inf}\right)}{T\eta_l\eta_g\tau} + 4\sigma_g^2\eta_l\left(\frac{\eta_g\nu}{m} + \frac{2L_s^2\eta_l\tau}{3} + L_s^2\tau\right)
$$

$$
+ \frac{80L_s^2\eta_l^2\tau^2\kappa'^2}{3} + \frac{48\eta_l\eta_g\nu(M-m)L_c^2}{\tau m(M-1)} \quad (123)
$$

With the small enough learning rate $\eta_l = 1/(\sqrt{T}\tau)$ and $\eta_g = \sqrt{\tau m}$ one can prove that

$$
\min_{t\in[T]}\mathbb{E}\left[\left\|\nabla\widetilde{F}(\mathbf{w}^{(t,0)})\right\|^2\right] \leq \frac{4\left(\widetilde{F}(\mathbf{w}^{(0,0)}) - \widetilde{F}_{\inf}\right) + 4\sigma_g^2\nu}{\sqrt{T\tau m}} + \frac{4\sigma_g^2 L_s^2}{\sqrt{T}} + \frac{8\sigma_g^2 L_s^2}{3\tau T}
$$

$$
+ \frac{80L_s^2\kappa'^2}{T} + \frac{48\nu(M-m)L_c^2\sqrt{\tau}}{\sqrt{T}m} \quad (124)
$$

$$
= \mathcal{O}\left(\frac{\sigma_g^2}{\sqrt{T\tau m}}\right) + \mathcal{O}\left(\frac{\sigma_g^2}{\tau T}\right) + \mathcal{O}\left(\frac{\kappa'^2}{T}\right) + \mathcal{O}\left(\frac{\sqrt{\tau}}{\sqrt{T m}}\right) \quad (125)
$$

completing the proof for Theorem 3.1 for partial client participation.

## D  Experiment Details and Additional Results

All experiments are conducted on clusters equipped with one NVIDIA TitanX GPU. The algorithms are implemented in PyTorch 1. 11. 0. All experiments are run with 3 different random seeds and the average performance with the standard deviation is shown. The code used for all experiments is included in the supplementary material.

### D.1  Experiment Details

For FMNIST, for the results in Fig. 4, Table 2, and Table 3, the data is partitioned into 5 clusters where 2 labels are assigned for each cluster with no labels overlapping across clusters. For the other FMNIST results and EMNIST, we use the Dirichlet distribution (Hsu et al., 2019) to partition the data with $\alpha = 0.5$, 0.05 respectively. Clients are randomly assigned to each cluster, and within each cluster, clients are homogeneously distributed with the assigned labels. For the Sent140 dataset, clients are naturally partitioned with their twitter IDs. The data of each client is partitioned to $60\% : 40\%$ for training and test data ratio unless mentioned otherwise.

| | Seen Clients | | | | | | | |
| --- | --- | --- | --- | --- | --- | --- | --- | --- |
| | FMNIST | | | | EMNIST | | | |
| | Byz=0.1 | | Byz=0.05 | | Byz=0.1 | | Byz=0.05 | |
| | Test Acc. | GM-Appeal | Test Acc. | GM-Appeal | Test Acc. | GM-Appeal | Test Acc. | GM-Appeal |
| MW-Fed | 17.24 $(\pm 2.35)$ | 0.01 $(\pm 0.0)$ | 21.28 $(\pm 1.79)$ | 0.02 $(\pm 0.0)$ | 15.83 $(\pm 1.52)$ | 0.004 $(\pm 0.0)$ | 22.22 $(\pm 0.63)$ | 0.008 $(\pm 0.001)$ |
| MAXFL | **69.42** $(\pm 2.87)$ | **0.35** $(\pm 0.05)$ | **70.60** $(\pm 2.76)$ | **0.42** $(\pm 0.03)$ | **52.74** $(\pm 0.44)$ | **0.20** $(\pm 0.01)$ | **56.10** $(\pm 0.77)$ | **0.23** $(\pm 0.01)$ |
| | Unseen Clients | | | | | | | |
| | FMNIST | | | | EMNIST | | | |
| | Byz=0.1 | | Byz=0.05 | | Byz=0.1 | | Byz=0.05 | |
| | Test Acc. | GM-Appeal | Test Acc. | GM-Appeal | Test Acc. | GM-Appeal | Test Acc. | GM-Appeal |
| MW-Fed | 18.45 $(\pm 2.81)$ | 0.01 $(\pm 0.0)$ | 21.91 $(\pm 3.81)$ | 0.01 $(\pm 0.0)$ | 17.03 $(\pm 0.21)$ | 0.005 $(\pm 0.0)$ | 22.23 $(\pm 0.63)$ | 0.003 $(\pm 0.0)$ |
| MAXFL | **69.75** $(\pm 3.66)$ | **0.39** $(\pm 0.01)$ | **71.11** $(\pm 1.47)$ | **0.46** $(\pm 0.01)$ | **53.82** $(\pm 0.09)$ | **0.26** $(\pm 0.02)$ | **55.10** $(\pm 0.78)$ | **0.28** $(\pm 0.01)$ |

Table 4: Byzantine clients are included in the total clients where they artificially report large losses to the server and add noise to their gradients. The ratio of the Byzantine clients is denoted as 'Byz'. We report the final avg. test accuracy and GM-Appeal across the seen and unseen clients where we train for 200 communication rounds. At the 10th round, clients flexibly opt-out or opt-in depending on whether the global model has met their requirements.

**Obtaining $\widehat{\mathbf{w}}_i$, $i \in [M]$ for MaxFL Results in Section 5.** In MAXFL, we use $\widehat{\mathbf{w}}_i$, $i \in [M]$ to calculate the aggregating weights (see Algorithm 1). For all experiments with MAXFL, we obtain $\widehat{\mathbf{w}}_i$, $i \in [M]$ at each client by each client taking 100 local SGD steps on its local dataset with its own separate local model before starting federated training. We use the same batch-size and learning rate used for the local training at clients done after we start the federated training (line 8-9 in Algorithm 1). The specific values are mentioned in the next paragraph.

**Local Training and Hyperparameters.** For all experiments, we do a grid search over the required hyperparameters to find the best performing ones. Specifically, we do a grid search over the learning rate: $\eta_l \eta_g \in \{0.1, 0.05, 0.01, 0.005, 0.001\}$, batchsize: $b \in \{32, 64, 128\}$, and local iterations: $\tau \in \{10, 30, 50\}$ to find the hyper-parameters with the highest test accuracy for each benchmark. For all benchmarks we use the best hyper-parameter for each benchmark after doing a grid search over feasible parameters referring to their source codes that are open-sourced. For a fair comparison across all benchmarks we do not use any learning rate decay or momentum.

**DNN Experiments.** For FMNIST and EMNIST, we train a deep multi-layer perceptron network with 2 hidden layers of units $[64, 30]$ with dropout after the first hidden layer where the input is the normalized flattened image and the output is consisted of 10 units each of one of the 0-9 labels. For Sent140, we train a deep multi-layer perceptron network with 3 hidden layers of units $[128, 86, 30]$ with pre-trained 200D average-pooled GloVe embedding (Pennington et al., 2014). The input is the embedded 200D vector and the output is a binary classifier determining whether the tweet sentiment is positive or negative with labels 0 and 1 respectively. All clients have at least 50 data samples. To demonstrate further heterogeneity across clients' data we perform label flipping to 30% of the clients that are uniformly sampled without replacement from the entire number of clients.

## D.2 Additional Experimental Results

**Robustness of MaxFL Against Byzantine Clients.** One may think that MAXFL may be perceptible to specific attacks from Byzantine clients that intentionally send a greater GM-Appeal gap to the server to gain a higher aggregation weight. To show MAXFL's robustness against such attacks we show in Table 4 the performance of MAXFL with Byzantine clients attacks which send higher losses to gain higher weights and then send Gaussian noise mixed gradients to the server. We compare with MW-Fed (Blum et al., 2021) which aims for incentivizing client participation by clients sending higher weights to the server and performing

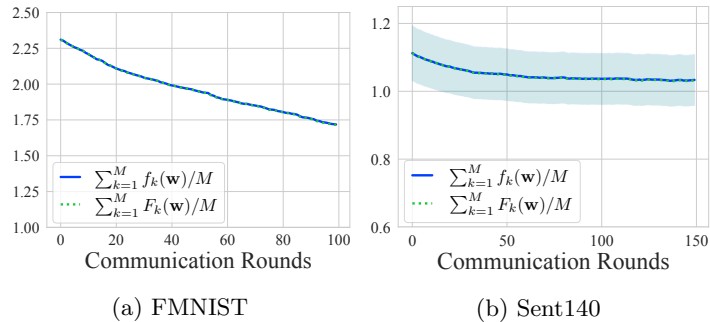

(a) FMNIST

(b) Sent140

Figure 6: Comparison of the average of the true local losses across all clients ($\sum_{k=1}^{M} f_k(\mathbf{w})/M$) and the empirical local losses across all clients ($\sum_{k=1}^{M} F_k(\mathbf{w})/M$) where the former is calculated on the test set and the latter is calculated on the training set for the global model $\mathbf{w}$. We show that the average of the true local losses is nearly identical to the average empirical local loss across all clients empirically validating our relaxation of replacing $f_k(\mathbf{w})$ with $F_k(\mathbf{w})$.

| | GM-Appeal | | | Preferred-Model Test Acc. | | |
|---|---|---|---|---|---|---|
| | $\tau_l = 50$ | $\tau_l = 100$ | $\tau_l = 150$ | $\tau_l = 50$ | $\tau_l = 100$ | $\tau_l = 150$ |
| FedAvg | 0.01 (±0.01) | 0.01 (±0.0) | 0.01 (±0.0) | 98.56 (±0.08) | 98.72 (±1.02) | 98.75 (±1.28) |
| FedProx | 0.01 (±0.0) | 0.01 (±0.0) | 0.01 (±0.01) | 98.56 (±1.15) | 98.72 (±1.08) | 98.75 (±1.05) |
| MaxFL | **0.55** (±0.0) | **0.55** (±0.0) | **0.55** (±0.0) | **98.64** (±1.03) | 98.77 (±1.01) | **98.77** (±1.01) |

Table 5: GM-Appeal and preferred-model test accuracy for varying number of local steps $\tau_l$ to obtain $\widehat{\mathbf{w}}_k, k \in [M]$ where $T = 200$ is the total number of communication rounds for training the global model.

more local updates. In Table 4 we see that for both high and low byzantine client ratios, MaxFL achieves only 1-5% lower test accuracy for seen and unseen clients compared to the case where there are no Byzantine clients in Table 1. This is due to our objective giving lower weight to those clients that give a too high GM-Appeal gap (see Fig. 3). Hence MaxFL disregards these clients that send artificially high GM-Appeal gaps. Note that it is indeed possible that attackers can devise different attacks, and in our work we consider one of them to show MaxFL's robustness. Specifically, our focus is on the case when some attackers try to modify their loss values in order to get higher weights for their updates.

**Ablation Study on $f_k(\mathbf{w}) \approx F_k(\mathbf{w})$.** One of the two key relaxations we use for MaxFL (see Section 2.1) is that we replace $f_k(\mathbf{w}) - f_k(\widehat{\mathbf{w}}_k)$ with $F_k(\mathbf{w}) - F_k(\widehat{\mathbf{w}}_k)$. In other words, we replace the true loss $f_k(\mathbf{w}) = \mathbb{E}_{\xi \sim \mathcal{D}_k}[\ell(\mathbf{w}, \xi)]$ with the empirical loss $F_k(\mathbf{w}) = \frac{1}{|\mathcal{B}_k|} \sum_{\xi \in \mathcal{B}_k} \ell(\mathbf{w}, \xi)$ for all clients $k \in [M]$. We have used the likely conjecture that the global model $\mathbf{w}$ is trained on the data of all clients, making it unlikely to overfit to the local data of any particular client, leading to $f_k(\mathbf{w}) \approx F_k(\mathbf{w})$. We show in Fig. 6 that this is indeed the case. For all DNN experiments, we show that the average true local loss across all clients, i.e., $\sum_{k=1}^{M} f_k(\mathbf{w})/M$ is nearly identical to the average empirical local loss across all clients, i.e., $\sum_{k=1}^{M} F_k(\mathbf{w})/M$ given the training of the global model $\mathbf{w}$ throughout the communication rounds. This empirically validates our relaxation of the true local losses to the empirical local losses.

**Ablation Study on the Number of Local Steps $\tau_l$ to train $\widehat{\mathbf{w}}_k, k \in [M]$.** We conduct an additional ablation study where we vary the number of local steps to obtain $\widehat{\mathbf{w}}_k, k \in [M]$ for clients as shown in Appendix D.2. Despite that a smaller number of local steps can lead to underfitting and a larger number of local steps can lead to overfitting, we show that all methods' GM-Appeals do not vary much by the different number of local steps used for training.

**Preferred-model Test Accuracy for the Local-Tuning Results in Table 3.** In Table 3, we have shown how MaxFL can largely increase the GM-Appeal compared to the other baselines even when jointly used with local-tuning. In Table 6, we show the corresponding preferred-model test accuracies. We show that for the seen clients that were active during training, MaxFL achieves at least the same or higher preferred-model test accuracy than the other methods for all the different datasets. Hence, the clients are able to also gain from MaxFL by achieving the highest accuracy in average with their preferred models (either

|  | Seen Clients | | Unseen Clients | |
|---|---|---|---|---|
|  | FMNIST | Sent140 | FMNIST | Sent140 |
| FedAvg | 99.37 (±0.24) | 55.71 (±0.46) | 99.50 (±0.02) | 58.79 (±0.67) |
| FedProx | 99.35 (±0.23) | 55.75 (±0.80) | **99.55** (±0.09) | 58.82 (±0.72) |
| PerFedAvg | 99.20 (±0.25) | 55.74 (±0.80) | 98.98 (±0.55) | 58.82 (±0.72) |
| MW-Fed | 99.27 (±0.39) | 55.06 (±0.38) | 99.47 (±0.08) | 57.36 (±0.71) |
| MAXFL | **99.40** (±0.30) | **55.82** (±0.82) | 99.50 (±0.02) | **58.88** (±0.77) |

Table 6: Preferred-model test accuracy with the locally-tuned models with 5 local steps from the final global model for seen clients' and unseen clients' test data (the corresponding GM-Appeal is in Table 3).

|  | GM-APPEAL | | Preferred-Model Test Acc. | |
|---|---|---|---|---|
|  | FMNIST | Sent140 | FMNIST | Sent140 |
| q-FFL ($q = 1$) | 0.03 (±0.01) | 0.09 (±0.06) | 99.24 (±0.05) | 53.10 (±2.63) |
| q-FFL ($q = 10$) | 0.0 (±0.0) | 0.09 (±0.0) | 98.90 (±0.01) | 52.71 (±1.40) |
| MAXFL | **0.55** (±0.0) | **0.41** (±0.07) | **99.29** (±0.03) | **53.93** (±1.87) |

Table 7: GM-APPEAL and preferred-model test accuracy for the seen clients' test data with the final global models trained via MAXFL and q-FFL (Li et al., 2019) which aims in improving fairness. The baseline q-FFL with large $q$, e.g. $q = 10$, emulates the behavior of another well-known algorithm named AFL (Mohri et al., 2019).

global model or solo-trained local model). For the unseen clients with FMNIST, FedProx achieves a slightly higher preferred-model test accuracy ($+0.05$) than MAXFL but with a much lower GM-Appeal of 0.46 (see Table 3) as MAXFL's GM-Appeal is 0.56. For the other datasets with unseen clients, MAXFL achieves at least the same or higher preferred-model test accuracy than the other methods. This demonstrates that MAXFL consistently largely improves the GM-Appeal compared to the other methods while losing very little, if any, in terms of the preferred-model test accuracy.

**Comparison with Algorithms for Fairness** Fair FL methods (Li et al., 2019; Mohri et al., 2019) aim in training a global model that yields small variance across the clients' test accuracies. These methods may satisfy the worst performing clients, but potentially at the cost of causing dissatisfaction from the best performing clients. We show in Table 7 that the common fair FL methods are indeed not effective in improving the overall clients' GM-APPEAL. We see that the fair FL methods achieve a GM-APPEAL lower than 0.01 for all datasets while MAXFL achieves at least 0.40 for all datasets. Moreover, the preferred-model test accuracy is also higher for MAXFL compared to the fair FL methods. This underwelming performance of fair FL methods in GM-APPEAL can be due to the fact that fair FL methods try to find the global model that performs well, in overall, over *all* clients which results in failing to satisfy *any* client.

**MaxFL's Theoretical Learning Rate Behavior for Fig. 1 (b)**. Here, we provide a plot of MAXFL's theoretical learning rate for the mean estimation example in Fig. 1(b) in Fig. 7 to show how the learning rate changes for different regions of the model. We show this plot as a proof of concept on the adaptive learning rate we discuss in Section 4. For the sigmoid function which is used for our MAXFL objective, using a global notion of smoothness can cause gradient descent to be too slow since global smoothness is determined by behavior at $w = 0$ where $w$ is the model. In this case, it is better to use a local estimate of smoothness in the flat regions where $|w| >> 0$. Recall that $\nabla^2 \sigma(w) = \sigma(w)(1 - \sigma(w))(1 - 2\sigma(w)) < \sigma(x)(1 - \sigma(w))$ and therefore setting the learning rate proportional to $\frac{1}{\sigma(w)(1-\sigma(w))}$ can increase the learning rate in flat regions where $\sigma(w)$ is close to 1 or 0. Following a similar argument, we can show that the learning rate in our objective should be proportional to $1/\left(\sum_{i=1}^{M} \sigma(F_i(w) - F_i(\hat{w}^*))(1 - \sigma(F_i(w) - F_i(\hat{w}^*)))\right)$.

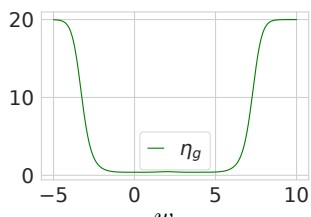

Figure 7: Behavior of Theoretical Learning Rate of MAXFL for the mean estimation example in Fig. 1(b). As expected from the theoretical learning rate formula, we see a higher learning rate in regions where the function is flat.

