# OpenReview forum: "Maximizing Global Model Appeal in Federated Learning"
_TMLR — Accepted by TMLR_

### Review · Reviewer_DCmo · 2023-12-17

**Summary Of Contributions:**

This paper considers a novel objective in federated learning, i.e., to maximize the total number of clients whose requirements are satisfied by the global model. The paper firstly proposes the novel metric of global model appeal, which measures the proportion of clients whose requirements are satisfied by the global model. Then, the proposed objective is relaxed in order to derive a practical objective and optimization algorithm to maximize the global model appeal. The convergence of the method is analyzed, and extensive experiments are performed to assess the advantage of the proposed metric and method.

**Audience:**

Yes

**Broader Impact Concerns:**

As I mentioned in the third point under Weaknesses above, the proposed method may cause fairness issues.

**Claims And Evidence:**

Yes

**Requested Changes:**

Most of the required changes are discussed under Weaknesses above. In addition,
- Theorem 2.1, should the $\gamma$ be $\gamma_G$?

**Strengths And Weaknesses:**

Strengths:
- The proposed metric of global model appeal is intuitive and practically useful, especially in problems in which the clients may opt out of the federation. It also provides a fresh perspective as to the goal of federated learning.
- Based on the proposed definition of global model appeal, the paper also applies natural relaxation of the definition, which leads to a natural algorithm which is able to efficiently maximize the objective of global model appeal.
- Although the toy example is very simple, it does allow for formalizing a number of interesting insights about the proposed global model appeal.
- The experimental results indeed show the empirical advantage of the proposed method.

Weaknesses:
- In your main experiments (such as Figure 1 and Table 1), do you consider the possibility of agents opting out of the federation? If yes, then this is different from the common experimental setting in federated learning, and needs to be clearly specified. Also, if you have indeed allowed the agents to opt out, how is this done exactly? An agent would opt out as long as its requirement is not satisfied in any iteration? In addition, if the answer is yes, have you tested the performance of the algorithm in the standard FL setting (no agents opting out)?
- In the toy example in Section 2.2, in the definition of $F_k(w)$, why is the second term $(\hat{\theta}_k - \theta_k)^2$ needed?
- The interpretation of the aggregated gradient in Section 3 is very interesting and provides insights and intuitions as to how the proposed algorithm works. However, I find the behavior of the algorithm when $F_k(\mathbf{w}) \gg \rho_k$ concerning. More specifically, the discussion at the top of page 6 suggests that in this case, the algorithm will simply "abandon" those agents for whom there is little hope to satisfy their requirements, i.e., give up helping them satisfy their requirements. I think this is likely to create problems regarding fairness. So, I think mitigation methods should be proposed or this issue should be given sufficient discussions.
- The theoretical results in Section 3.1 can be presented better. For example, I think it would be helpful to clearly discuss what are the insights drawn from the theoretical results, and what how do these theoretical results differ from those of classical FedAvg.
- On page 9, it is mentioned that "We perform grid search for hyperparameter tuning for all baselines and choose the best performing ones". Is this a common practice? I imagine in practice, it may make more sense to use a single set of hyperparameters in all new applications, or to use a validation set to select hyperparameters and then test the results using a separate test set.
- In the experiments of "Local Tuning for Personalization", have you compared the performance using local fine-tuning with the performance without it (no personalization)?

---

> ### Author Response · Authors · 2024-01-20
> **Response to Reviewer DCmo #1**
>
> Thanks for your comments! We address the reviewer's concerns below:
>
> **[Possibility of Agents Opting Out in Experiments].** Yes, in our experiments we do include scenarios in which clients can opt out if their GM-Appeal is not satisfied. Specifically, as mentioned in the top paragraph of page 7, we assume that all clients are available for training if selected for a few initial training rounds ($5$ to $10$ communication rounds) for warm-up. After these warm-up rounds, the server can only select clients from the pool of clients which find the global model appealing (i.e. their requirement is satisfied) in that particular round. We demonstrate this extension of MaxFL with appeal-based flexible client participation in Table 1 and Table 2. We have also included experiments without flexible client participation in Figure 4 showing that even in this case MaxFL maximizes the GM-Appeal and preferred-model test accuracy.
>
>
> **[The term $(\hat{\theta}_k-\theta_k)^2$].** We add the second term $(\hat{\theta}_k-\theta_k)^2$ to ensure that
> that $\rho_k = F_k(\hat{\theta}_k)  = (\hat{\theta}_k-\theta_k)^2 > 0$. Otherwise, we would have $\rho_k = 0$ for all $k \in [2]$ and the GM-Appeal for any method will always be $0$. Hope this clarifies your concern.
>
> **[Concerns on Fairness].** Indeed, there can be fairness concerns in MaxFL behavior exhibited in Figure 3 where in order to retain the maximum number of satisfied clients, MaxFL gives lower weights to those clients that seem as outliers to the majority data distribution. While fairness is orthogonal to MaxFL's objective, we do include experiments and discussions on fairness in Section 4.2 and Table 8 in Appendix D.3. We show that while fair FL methods aim in training a global model that yields small variance across the clients’ test accuracies, these methods may satisfy the worst performing clients, but potentially at the cost of causing dissatisfaction for the best performing clients. We show in Table 8 that common fair FL methods are indeed not effective in improving the overall clients’ GM-Appeal. We see that the fair FL methods achieve a GM-Appeal lower than 0.01 for all datasets while MaxFL achieves at least 0.40 for all datasets. Moreover, the preferred-model test accuracy is also higher for MaxFL compared to the fair FL methods. This underwhelming performance of fair FL methods in GM-Appeal can be due to the fact that fair FL methods try to find the global model that performs well, in overall, over all clients which results in failing to satisfy any client. We believe it is not MaxFL's objective to guarantee fairness but rather obtain the maximum number of clients that are satisfied, and asking MaxFL to mitigate fairness issues as well is out of the scope of this paper.
>
>
> **[Theoretical Results in Section 3.1].** Thank you for your suggestion. We will revise the theoretical result section in our updated version to summarize the main insight which is: i) theoretically, MaxFL's optimal solution for mean estimation guarantees a higher GM-Appeal than the standard FedAvg's optimal solution by adapting to data heterogeneity.
>
> **[Grid Search for Hyperparameter Tuning].** Different optimizers and tasks may require different sets of optimal hyperparameters, and therefore to make a fair comparison across baselines and tasks we performed a grid search on the set of hyperparameters for each.
>
> **[Local Fine-tuning without Personalization].** We are a bit confused of the comment on showing the performance of local fine-tuning without personalization. In the experiments of "Local Tuning for Personalization", local tuning is considered personalization itself. Can the reviewer clarify what they mean by local fine-tuning without personalization?
>
> **[Theorem 2.1, should the $\gamma$ be $\gamma_G$?].** It should be $\gamma$ where $\gamma$ is defined as $\gamma := \nu^2/N$ (variance of local mean estimation). Intuitively, if $\gamma$ is small, it implies that each client can accurately estimate its local mean and has little incentive to use the global model leading to a $\exp(-1/\gamma^2)$ term in our GM-Appeal lower bound. Please see our proof in Appendix B for more details.
>
> We hope we have clarified the concerns of the reviewer. Please let us know of any remaining concerns!

---

> > ### Comment · Reviewer_DCmo · 2024-02-21
> > **Thank you for the response**
> >
> > I'd like to thank the authors for the response, which clarified most of my concerns. The explanation w.r.t. fairness makes sense to me. Fairness and performance can be seen as a trade-off, and it makes sense that putting too much emphasis on fairness can hurt the overall performance (overall accuracy or GM-Appeal). I think adding these discussions to the paper will be very helpful.
> >
> > Regarding the question about local fine-tuning with personalization, what I in fact meant is whether you have compared local fine-tuning with not performing local fine-tuning at all. But this is not a major concern, what I had in mind was just a sanity check to verify whether personalization actually helps in the experiments in this paper.

---

### Review · Reviewer_xZWJ · 2023-12-25

**Summary Of Contributions:**

This paper introduce a novel objective called Global Model Appeal (GM-Appeal), which aims at training a global model that can satisfy the requirements of the most number of local clients. To maximize the proposed GM-Appeal, the author propose a framework called MaxFL, which are shown to be able to perform well in the new objective. In addition, convergence analysis are provided for the proposed method.

**Audience:**

Yes

**Broader Impact Concerns:**

Broader impact is properly addressed

**Claims And Evidence:**

Yes

**Requested Changes:**

Please see the "weakness" section.

**Strengths And Weaknesses:**

Strength:
1) The proposed metric is well-motivated and has real world application potentials. This proposed new framework can inspire the community to rethink about the traditional global loss minimization objective.
2) Extensive numerical validation are provided from various aspects to show effectiveness of the proposed method.
3) Theoretical analysis are provided for the method.

Weakness:
1) Since the proposed objective resembles a "hard" variation of q-FFL, I suggest the author provide more discussion to compare between q-FFL and the proposed GM-Appeal.

---

> ### Author Response · Authors · 2024-01-20
> **Response to Reviewer xZWJ #1**
>
> Thanks for your comments!
>
> **[Comparison betweeen q-FFL and GM-Appeal].** We disagree that MaxFL is a hard variation of q-FFL. Can the reviewer clarify why they think this is the case? MaxFL does not give higher weights to those clients  that have higher local loss like q-FFL does, and rather considers the GM-Appeal gap and gives higher weight to those clients that have close GM-Appeal gap to the majority of clients. In addition, MaxFL actually gives lower weights to those clients with high GM-Appeal gap or low GM-Appeal gap (see Figure 3). Nevertheless, we do include experiments and discussions on fairness in Section 4.2 and Table 8 in Appendix D.3. We show that while fair FL methods aim in training a global model that yields small variance across the clients’ test accuracies, these methods may satisfy the worst performing clients, but potentially at the cost of causing dissatisfaction from the best performing clients. We show in Table 8 that the common fair FL methods are indeed not effective in improving the overall clients’ GM-Appeal which is MaxFL's main objective.
>
> We hope we have clarified the reviewer's concerns! Please let us know if there are any further questions.

---

### Review · Reviewer_E382 · 2024-01-11

**Summary Of Contributions:**

The authors propose the so-called Global Model Appeal (GM-Appeal) as an alternative to the standard FL objective. This quantity measures the ratio of clients finding the current global model $w$ "appealing", i.e., the ratio of clients such that the value of true loss function $f_k(w)$ of client $k$ is smaller than some predefined threshold $\rho_k$. The authors also provide a smooth relaxation of the proposed formulation (via replacing the indicator function with a sigmoid and the true loss with an empirical one) and design a new algorithm (called MaxFL) for minimizing the resulting objective. This algorithm can be seen as FedAvg for the relaxed GM-Appeal minimization with a stepsize that adapts to the weights of aggregation. The authors also analyze the proposed method for non-convex problems with smooth, Lipschitz-continuous objective under bounded variance and bounded gradient dissimilarity assumptions. MaxFL was also tested in numerical experiments and showed better GM-Appeal and test accuracy than competitors on image classification tasks (FMNIST, EMNIST, Sent140).

**Audience:**

Yes

**Broader Impact Concerns:**

The authors provide enough details on a broader impact.

**Claims And Evidence:**

No

**Requested Changes:**

## Questions and suggestions for improvement

1. Can the benefits of maximizing GM-Appeal be formalized and justified theoretically? For example, one can try to come up with an example when GM-Appeal maximization is provably beneficial in the case when new clients join with a data similar to the seen clients. If it is problematic to show theoretical benefits of GM-Appeal maximization, then more substantial numerical justification is required.

2. How well does problem (3) approximate problem (2)?

3. In the toy example, $\rho_k$ requires knowing $\theta_k$. In this case, the mean is already known for each client.

4. The experiment with Byzantine attackers is confusing. In this experiment, the authors assume that Byzantine workers send too large GM-Appeal gap. However, it is unclear why it is the worst possible scenario. For example, Byzantine workers can send zero as a "gap" and send arbitrary vectors as $\Delta w_k^{(t,0)}$, e.g., these vectors can have arbitrary large norms. In this case, MaxFL can be broken by Byzantines in 1 communication round. Therefore, in the current shape the algorithm cannot resist Byzantine attacks.

5. On page 6, the authors claim that MaxFL allows "partial client availability". This statement is inaccurate since the authors use uniform client sampling, which implicitly relies on the fact that all clients are available at any time (since every client can be sampled with non-zero probability).

6. The pseudocode of the method should be provided in the main part of the paper. Otherwise, it is hard to understand how the method works and what are some of the quantities stand for (in particular, it is mentioned only in the pseudocode that $T$ stands for the number of communication rounds).

7. On page, the authors mention that the optimal learning rate for MaxFL is $\widetilde{\eta} = 1/\widetilde{L}_s$. Is there any proof of optimality of this stepsize? Did the authors try to use other stepsizes (e.g., constant ones, without weights) in MaxFL for the experiments?

8. There is no complete formulation of Theorem 3.1 in the appendix (though the authors claim that they provide it in the appendix). In particular, complete formulas for stepsizes are required together with the requirements on $T$, i.e., how large $T$ should be? Moreover, the current formula for $\eta_l$ and the third and fourth terms in RHS of (5) have issues with a physical dimension: $\eta_l$ should have the same physical dimension as $1/L_s$ while $T,\tau,m$ have void physical dimension.

9. Why does the inequality for $\widetilde{\eta}^{(t,0)}$ between formulas (95) and (96) hold?

## Minor comments

1. Definition 1: the notation for GM-Appeal does not reflect the fact that it depends on $w$ and $\rho_k$. I suggest to replace GM-Appeal with GM-appeal($w, \lbrace\rho_k\rbrace_{k\in [M]}$).

2. Formula (2): usually $\text{sign}(x)$ denotes a sign of $x \in \mathbb{R}$, i.e., $-1,0,+1$. The function that authors denote as $\text{sign}(x)$ is usually denoted as $[x]_+$.

3. Page 6: wrong sign in the definition of $\Delta w_k^{(t,0)}$.

4. I think there is a typo in (55) in the calculation of the Hessian, though starting from (58) the derivation is correct.

5. In the derivation of (91), the authors use (47) + (49) + (52) instead of just the statement of Lemma C.1. So, it is better to adjust the statement of Lemma C.1.

6. The first line of page 26: should be $\eta_l \eta_g \leq 1 / (4\tau M L_s)$.

**Strengths And Weaknesses:**

### Strengths

S1. The proposed formulation is new and shows good performance in simple experiments.

S2. Numerical results show that MaxFL works well even when some clients decide to leave/join.

### Weaknesses

W1. The benefits of considering GM-Appeal maximization are not formalized and not justified theoretically. The authors explain the intuition and motivation behind the proposed formulation but do not show formal benefits of its consideration.

W2. The authors do not provide a formal result on how problems (2) and (3) are related.

W3. Theorem 3.1 relies on many assumptions. In particular, the authors assume Lipschitzness and smoothness of $F_k$. Typically, for the analysis of optimization methods and, in particular, FL algorithms it is sufficient to assume either Lipschitzness or smoothness.

---

> ### Author Response · Authors · 2024-01-20
> **Response to Reviewer E382 #1**
>
> Thanks for your comments! We address the reviewer's comments below:
>
> **[More Substantial Numerical Justification].** We believe we have already provided thorough numerical justification of our proposed MaxFL including a wide variety of experiments across three different tasks: i) server attaining more participating clients to select from (increasing GM-Appeal), ii) showing that with higher GM-Appeal the global model has a higher chance to have a good performance on unseen clients (as the reviewer has asked for justification), iii) MaxFL's robustness against byzantine clients, iv) MaxFL's performance when combined with personalization. Aside from this, we also provide additional experimental results in the appendix to perform an ablation study on the approximation of the true loss $f_k(\mathbf{w})$ and empirical loss $F_k(\mathbf{w})$, ablation study on the number of local steps to obtain the client's local model $\hat{\mathbf{w}}_k$, and also comparison with algorithms for fairness. Lastly, we also provide theoretical insights for the mean estimation problem which show that MaxFL can guarantee more number of satisfied clients than FedAvg. We politely disagree that our work lacks in numerical justification of the effectiveness of MaxFL to maximize GM-Appeal.
>
> **[Approximation from (2) to (3)].** There are two main assumptions we make to approximate (2) to (3). The first is replacing the true local loss $f_k(\mathbf{w})$ to the empirical local loss $F_k(\mathbf{w})$, which we show in Appendix D.3 that indeed this approximation is reasonable since $f_k(\mathbf{w})\simeq F_k(\mathbf{w})$ in most cases. The second assumption is replacing the sign function to the sigmoid function. As mentioned in detail in Section 2.2 and Figure 3, the approximation with the Sigmoid function is essential to the success of MaxFL able to maximize the GM-Appeal since it allows the server to assign higher weights to those clients that are not outliers in terms of the values of the GM-Appeal gap (i.e., $F_k(\mathbf{w})-\rho_k$). Through both theoretical analysis and experimental validation we believe we have thoroughly justified the approximation to the Sigmoid function.
>
> **[Client knowing their mean for Toy Example].** Note that the purpose of the mean estimation toy problem is purely to give theoretical insights and justification of the MaxFL objective and its approximation to the Sigmoid. As such, we utilize the information of the true mean $\theta_k$ when defining $F_k(w)$ which gives us $\rho_k = F(\widehat{w}_k) = (\hat{\theta}_k -\theta_k)^2$. This is primarily done to ensure that $\rho_k > 0$, else if $\rho_k = 0$ then the global model will never be appealing to client $k$. This is not an issue for practical neural network tasks since we typically have $F(\widehat{\mathbf{w}}_k) > 0$ and we do not make use of the true model in our experiments.
>
>
> **[Other Possible Byzantine Attacks].** It is indeed possible that attackers can devise different attacks, and in our work we consider one of them to show MaxFL's robustness. Specifically, our focus is on the case when some attackers try to modify their loss values in order to get higher weights for their updates. The issue of the updates having arbitrary large norms can be mitigated by enforcing update norm clipping or normalization [1]; however we did not explore this in our current work.
>
>
> **[Partial Client Participation].** We will rephrase the term partial client availability to partial client participation in our revised version.
>
> **[Pseudo Code].** Thanks for the suggestion! We will move the pseudo-code to the main paper in our revised version and also clarify earlier on what $T$ stands for.
>
> **[Proof of Optimality of the Learning rates].** Yes we give the validation of the optimality of the learning rate in Appendix C where the proof for the convergence of MaxFL leads us to this optimal learning rate. We also do a deeper dive into the behavior of MaxFL's theoretical learning rate behavior in Figure 7 in Appendix D.3 to show the intuition of why such learning rate can be optimal. In practice, it is difficult to know the Lipschitz constants of the local objectives and we perform a grid search for the learning rate settings for MaxFL.
>
> **[Complete Formulation of Theorem 3.1].** We do include a complete formulation of our Theorem in the Appendix in eq. (100) and eq. (125) for full and partial client participation respectively. We will include them in a separate theorem in the Appendix for better reading in our revised version. There is no restriction we make throughout our proof on how large $T$ should be. It can be seen that as $T\rightarrow\infty$, the terms in our convergence bounds goes to $0$.

---

> > ### Author Response · Authors · 2024-01-20
> > **Response to Reviewer E382 #2**
> >
> > **[(95) to (96)].** The inequality for $\tilde{\eta}$ holds due to our setting of the learning rate to $\tilde{\eta}=1/\tilde{L}_s$ $=M\eta / ( \sum{k\in[M]} q_k(\mathbf{w})+\epsilon)$, as mentioned in page 6 in our main paper.
> >
> >
> > <References>
> >
> > [1] Das, Rudrajit, Abolfazl Hashemi, and Inderjit S. Dhillon. "Differentially private federated learning with normalized updates." OPT 2022: Optimization for Machine Learning (NeurIPS 2022 Workshop). 2022.
> >
> >
> > We hope we have clarified the reviewer's concerns and look forward to their feedback!

---

> > ### Comment · Reviewer_E382 · 2024-02-15
> > **Response to authors' replies**
> >
> > I thank the authors for the replies that resolved some of my questions and concerns. However, I still have several concerns.
> >
> > **On W1.** I agree that the provided numerical results illustrate some potential benefits of the usage of GM-appeal maximization. However, since the provided theory does not formally explain why (and when) maximizing GM-appeal is better than standard ERM minimization, *either* additional experimental studies on more complex problems (different datasets, different models) *or* improved theoretical results are required.
> >
> > **On W2.** The empirical study given in Appendix D.3 justifies this hypothesis for two considered tasks (for a particular number of clients, data, and models). However, without formal theoretical guarantees, the same criticism is valid as in the above point: either additional experimental validation is needed or some theoretical guarantees are required (even in some simple case). This is important because it will help to understand the limitations of the approach better.
> >
> > **On Byzantine attacks.** Since the algorithm is not robust to any Byzantine attacks, the claims in the paper should be adjusted accordingly. In particular, in the current shape, one cannot claim that the method is Byzantine-robust. Moreover, the usage of clipping/normalization of the updates does not fix the issue: Byzantine workers can still send uncorrelated weights $q_k$ and model updates pointing in the wrong direction. This will create an irreducible shift in the model update.
> >
> > **On Theorem 3.1.** Let me clarify my comment. The theorem states that "for sufficiently large $T$" formula (5) holds. How large $T$ should be for this theorem to hold? Also I want to point out to my comment about the problems with physical dimensions for $\eta_l$ and the third and fourth terms in RHS of (5).
> >
> > **(95) to (96).** Could the authors provide a complete derivation? How $\eta_l$, $\eta_g$ and $\eta$ are related?

---

### Decision · Action_Editor_1cxY · 2024-02-21

**Recommendation:** Accept with minor revision

**Comment:**

This paper proposes a new objective for federated learning, which is to optimize a notion of "global model appeal" to the clients rather than the average loss over clients. This idea is new, significantly different from previous proposals of alternative objective functions, and resonates with the need of providing incentives to convince clients to join an FL process.

The proposed formulation is validated by some theoretical and empirical results. While I agree with reviewer E382 that these results could be improved/broadened, I side with the other two reviewers that the current results are sufficient to validate the usefulness of the proposed idea, which I believe has the potential to be refined in subsequent work.

For these reasons, I recommend the paper to be accepted with minor revision. The requested changes are as follows:
- The claim that the proposed approach is "robust to Byzantine clients", while not central to the paper, is not supported by sufficient evidence, as highlighted by reviewer E382. Indeed, Byzantine clients can behave arbitrarily, hence such a robustness property must be formally established. What the authors provide is evidence that their approach is robust to a particular attack (but better attacks may exist). I thus ask the authors to tone down their claim, e.g., by stating that their approach exhibits better robustness to some attacks.
- In his/her last message, reviewer E382 asked for clarifications about Theorem 3.1 and its proof. Please make sure you add these details.

**Audience:**

This paper proposes a new objective for federated learning, which is of broad interest to TMLR's audience.

**Claims And Evidence:**

The main claims are supported by accurate theoretical and empirical evidence. One minor claim regarding robustness to Byzantine clients should be toned down, see below for details.

---

> ### Author Response · Authors · 2024-03-09
> **Thank you for your comments**
>
> We thank all the reviewers and editors for the constructive feedback and comments on our work! We have revised the paper according to the reviewers' and editor's comments including the following main changes:
> - The claim on MaxFL being robust against Byzantine clients has been softened and clarified by specifying which kind of attacks MaxFL is robust against and acknowledging that there can be attacks which MaxFL may not be as robust against.
> - We reflected reviewer E382's comments on changing the term partial client 'availability' to 'participation', and moving the pseudo-code to the main paper, and clarifying parts of our Theorem 3.1 that were discussed.
> - We reflected reviewer DCmo's comment to highlight our theoretical contribution in Section 3.1.
>
> We appreciate all the time the reviewers and editors have taken to review this paper.
>
> -- Authors